# Major restructuring of marine plankton assemblages under global warming

Fabio Benedetti [1✉], Meike Vogt[1], Urs Hofmann Elizondo [1], Damiano Righetti[1], Niklaus E. Zimmermann [2,3] & Nicolas Gruber [1]

Marine phytoplankton and zooplankton form the basis of the ocean's food-web, yet the impacts of climate change on their biodiversity are poorly understood. Here, we use an ensemble of species distribution models for a total of 336 phytoplankton and 524 zoo-plankton species to determine their present and future habitat suitability patterns. For the end of this century, under a high emission scenario, we find an overall increase in plankton species richness driven by ocean warming, and a poleward shift of the species' distributions at a median speed of 35 km/decade. Phytoplankton species richness is projected to increase by more than 16% over most regions except for the Arctic Ocean. In contrast, zooplankton richness is projected to slightly decline in the tropics, but to increase strongly in temperate to subpolar latitudes. In these latitudes, nearly 40% of the phytoplankton and zooplankton assemblages are replaced by poleward shifting species. This implies that climate change threatens the contribution of plankton communities to plankton-mediated ecosystem services such as biological carbon sequestration.

[1] Environmental Physics, Institute of Biogeochemistry and Pollutant Dynamics, ETH Zürich, Zürich, Switzerland. [2] Dynamic Macroecology, Swiss Federal Research Institute WSL, Birmensdorf, Switzerland. [3] Department of Environmental Systems Science, ETH Zurich, Zürich, Switzerland. ✉email: fabio.benedetti@usys.ethz.ch

The species diversity of marine plankton governs some of the most important marine ecosystem services[1–4]. In the sun-lit layers of the oceans, photoautotrophic phytoplankton are responsible for about 50% of Earth's annual net primary production[5]. Phytoplankton are mainly grazed by the heterotrophic zooplankton, which in turn sustain global fisheries production[1,2]. Together, these two trophic levels drive the biological carbon pump, a key determinant of the ocean-atmosphere balance of $CO_2$[2,3,6,7]. The diversity of phyto- and zooplankton has been shown to be a key modulator of this pump[4,8]. Similarly, the diversity and size structure of the zooplankton mediate the recruitment of economically important fishes[1,6,8]. The majority of studies indicate that the diversity of phyto- and zooplankton is largely controlled by climate[9–11], with temperature being the main driver[9–15]. Warm temperatures promote species diversity by enhancing speciation, metabolic rates and selecting for a higher number of species[11–13,15]. However, ocean warming forces species to shift their distribution ranges poleward to track their optimal thermal habitats[16,17] and such shifts have weakened the strength of the biological carbon pump over the past 55 years in the North Atlantic[6]. Furthermore, future global warming is expected to trigger species extirpations by pushing species beyond their thermal limits and by restructuring community composition[18,19], both associated with potentially deleterious consequences for the functioning of marine food-webs and biogeochemical cycles. This is because these processes are mediated by species-level functional traits and biological interactions[3,20,21]. Yet, the extent to which plankton diversity and associated ecosystem functions might respond to future warming remains poorly understood across clades and trophic levels. Earth system models (ESMs) with embedded marine ecosystem models cannot be used to this effect, as they still insufficiently resolve the variety of traits and functions performed by the numerous species of plankton[22]. Historically, field observations have been too sparse to undertake empirical model-based projections of global diversity for the various plankton groups[10,12,13]. Thus, previous projections of future plankton species diversity were based on either virtual taxa[18] or on a spatially and temporally very limited set of observations[10]. Consequently, the extent to which global phyto- and zooplankton diversity might be affected by future climate change remains unclear[23].

We address these limitations by modeling the monthly and mean annual diversity patterns stemming from the distribution of 860 plankton species (336 phytoplankton, 524 zooplankton) spanning 13 phyla, 71 orders, and 324 genera through an ensemble approach based on species distribution models (SDMs[24]). The considered species cover a wide range of traits and functions, representing ten major plankton functional groups (PFGs; three phytoplankton and seven zooplankton groups, see "Methods"). We compile the species occurrence records ($n = 934,696$) from various data sources and aggregate them onto a monthly resolved 1° x 1° grid, excluding observations from regions where the seafloor is shallower than 200 m. We match these binned open ocean records with observation-based climatologies of environmental predictors (temperature, dissolved oxygen concentration, solar irradiance, macronutrients concentration, chlorophyll a concentration) that reflect the climatic and biogeochemical conditions of the surface open ocean. Four types of SDMs (generalized linear models, generalized additive models, artificial neural networks, and random forests[24]) are fitted to model the species' current environmental habitat suitability patterns. For each SDM, we use four alternative pools of predictors. Assuming niche conservatism, we project each of the 16 resulting habitat suitability models into the future using outputs from five ESMs belonging to the Coupled Model Intercomparison Project 5 (CMIP5[25],) that were forced by the Representative

Concentration Pathway 8.5 (RCP8.5[25],) scenario of high greenhouse gas concentrations. To this end, we first compute the modeled monthly climatologies of the selected predictors for the 2012–2031 and 2081–2100 periods, and derive the future monthly anomalies from the differences between these two time periods. These anomalies are added to the observation-based monthly climatologies (i.e., those used to train the SDMs) to estimate the future environmental conditions of the ocean[26], and project the SDMs into these future conditions. Finally, we estimate mean annual present and future alpha diversity (species richness; SR) and beta diversity (species turnover through time) patterns for both trophic levels for each cell from the SDM ensembles. SR ensembles are estimated as the sum of all species' habitat suitability patterns averaged across all 80 possible combinations (hereinafter called ensemble members) of SDMs ($n = 4$), ESMs ($n = 5$), and predictor pools ($n = 4$). To assess the uncertainties of our diversity projections based on the ensemble members, we compute the interquartile range of the 80 ensemble members SR projections. We calculate species turnover as the change in mean annual species composition between present and future time based on Jaccard's dissimilarity index and by decomposing this total turnover into the true species turnover (ST, also known as species replacement) and the nestedness (SR change) components[27]. Numerous tests are conducted to ensure the robustness of the results with regard to the spatially and temporally highly uneven sampling effort as well as with regard to the relative role of different predictors.

Our ensemble of models project phytoplankton SR to increase by more than 16% in most basins except the Arctic Ocean and zooplankton SR to strongly increase in temperate to subpolar latitudes (+24%) but to slightly decline in the tropics (−4%). In the temperate and subpolar latitudes, turnover rates as high as 40% are predicted for the mean annual phyto- and zooplankton assemblages compositions, which implies that future climate change threatens the plankton-mediated ecosystem services provided by the ocean in these regions.

## Results and discussion

**Contemporary latitudinal species diversity gradients.** For the present time (2012–2031), we predict a strong latitudinal diversity gradient with annual mean plankton SR decreasing from the equator to the poles (Fig. 1a, b). This pattern of plankton SR emerges from overlaying the SR of phytoplankton (Fig. 1c, d) and zooplankton (Fig. 1e, f), with the latter contributing 61% to the total number of species modeled. Although both groups display conventional latitudinal diversity gradients, their SR maxima are not collocated. Phytoplankton SR peaks near the equator with maxima in tropical upwelling regions, whereas zooplankton SR peaks in the subtropics (20°–30°) and slightly decreases toward the tropical upwelling regions. The broad maximum in zooplankton SR in the low latitudes and its rapid decrease with latitude matches the SR patterns previously reported for several planktonic and non-planktonic oceanic taxa[12,13,15]. The diversity of the ten PFGs considered is also maximal in the low latitudes and decreases toward the poles, yet their respective SR maxima are distinct (Supplementary Note 6). The SR patterns of diatoms, dinoflagellates, haptophytes, chaetognaths and pteropods peak in the tropical band (0–30°) with some inter-PFGs variability (e.g., haptophyte SR decreases where diatom and dinoflagellate SR increases). On the other hand, the SR of the other zooplankton PFGs (copepod, euphausiid or jellyfish) peaks in the subtropics. The tropical dip in zooplankton SR is not a result of a spatial sampling bias[28], since a rarefaction approach, where we ensured a more even sampling effort across latitudes, provided very similar zooplankton SR patterns (Supplementary Note 7). The latitudinal

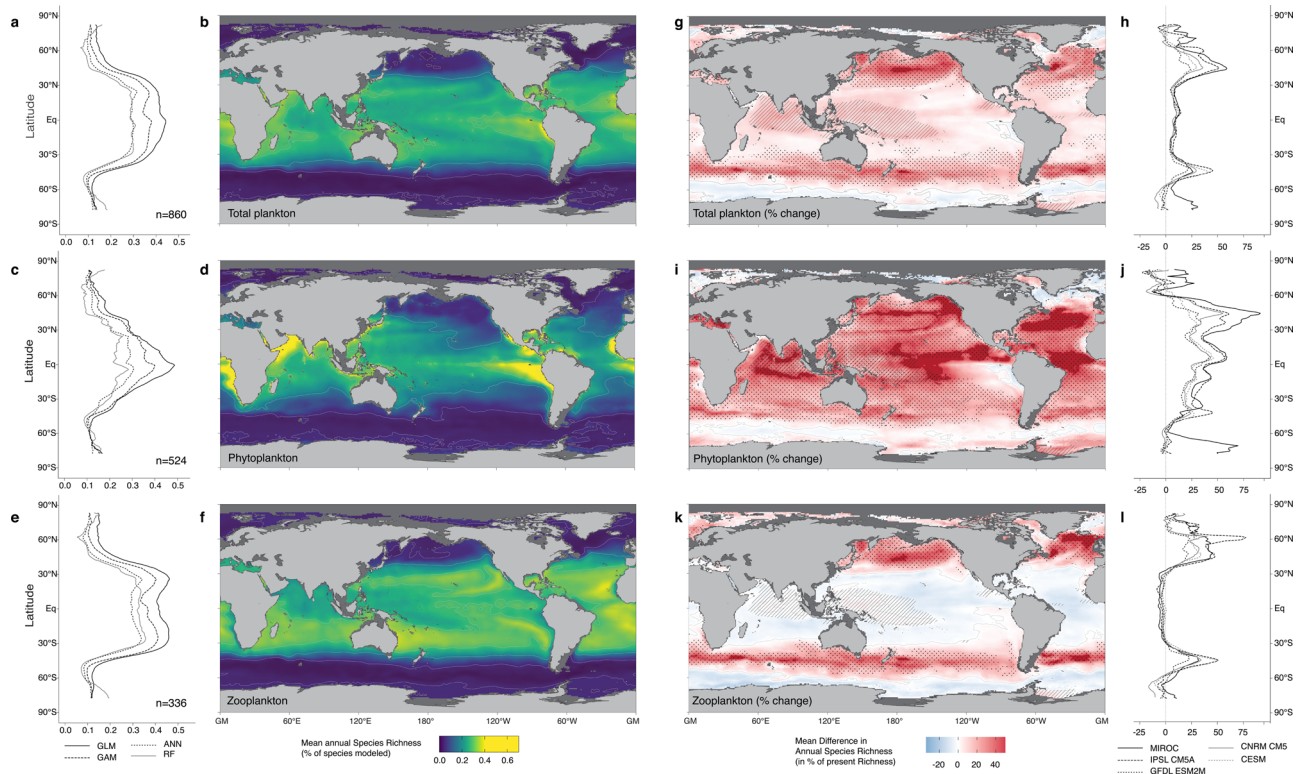

**Fig. 1 Global patterns of annual mean species richness (in % of species modeled) of plankton (first row), phytoplankton (second row) and zooplankton (third row) in the contemporary surface ocean (left two columns) and their projected changes for the 2081–2100 period under the representative concentration pathway RCP8.5 scenario. a, b** The annual mean species richness for all 860 phyto- and zooplankton species modeled. Shown is the mean richness across all 16 ensemble members (4 species distribution models and 4 predictor pools), **c, d** same as **a–b** but for the 336 modeled phytoplankton species. **e, f** same as **a**, **b**, but for the 524 zooplankton species. **a–e** show the zonal averages separately for each of the four statistical models. **g, h** the percentage difference in species richness induced by future climate change for all plankton species. Shown is the mean across all 80 ensemble members (4 species distribution models, 4 predictor pools, 5 earth system models. **l**, **j** same as **g**, **h** but for phytoplankton. **k**, **l** same as **g**, **h** but for zooplankton. **h–l** show the corresponding zonal averages in percentage difference for each of the 5 earth system models. The gray contour lines represent the isopleths for 50 units of mean annual species richness in **b–f** and for 25% of species richness difference in **g–k**. Regions in darker red highlight gains in annual SR larger than 50%. Stippling in **g–k** marks the regions where at least 90% of the 80 ensemble agree on the direction of the change. Hatching marks the region where non-analog conditions emerge in the future ocean on an annual scale. Total sample size is N = 35,023 grid cells.

SR gradients modeled for the ten PFGs match previous global and/or regional observations (Supplementary Note 6), which gives us confidence that our modeling framework captures the large-scale drivers of marine plankton diversity.

**Future changes in species diversity.** For the end of the century (2081–2100; Fig. 1g–l), we project a significant global median increase in mean annual plankton SR of 5% (1.9–9.6%, $p < 10^{-15}$; 25th–75th; Wilcoxon–Mann–Whitney test between median levels of baseline and future annual SR of each ensemble member). This global increase is driven by a strong gain in SR projected for the temperate to subpolar latitudes between 40° and 55° (+22%; 17–62%; $p < 10^{-15}$) and is offset by smaller gains projected for the tropical latitudinal band (30°S–30°N; +4%; 0–32%; $p < 10^{-8}$). The projections made for total plankton mask strong differences between the SR response of phytoplankton (Fig. 1I, j) and zooplankton (Fig. 1k, l) to future climate change. Phytoplankton SR increases globally by 16% (11–22%; $p < 10^{-15}$), with an even stronger enhancement in the tropical band (+21%; 12–100%; $p < 10^{-15}$) and a more modest one at temperate and subpolar latitudes (+13%; 10–38%; $p < 10^{-15}$). This is partially offset by a modest decrease in SR north of 70°N ($-11\%$; $-17$ to $+24\%$; $p < 10^{-9}$). In contrast, global mean zooplankton SR shows little changes with a near zero median response of +0.4%. However, this is the result of a compensation between regionally strongly

differing trends. In the temperate to subpolar latitudes (40°–55°) zooplankton SR is projected to increase by 24% (19–69%; $p < 10^{-15}$), while it is projected to slightly decrease in the tropical band ($-4\%$; $-6$ to $+7\%$; $p < 10^{-6}$) and in the Southern Ocean ($-3\%$; $-9$ to $+5\%$; $p < 10^{-12}$).

The predicted changes in zooplankton SR match the prevailing view that ocean warming increases diversity in cold temperate regions[16,18] through the poleward shifts of marine ectotherms tracking their thermal optima[2,16–18]. However, our global study demonstrates that the SR of the two trophic levels might experience different trajectories[10,18,19]. Differences in SR responses are also marked between the PFGs (Supplementary Note 6 and 9). While the SR of diatoms, dinoflagellates and haptophytes decreases in the Arctic and increases in temperate latitudes, their responses diverge in the tropics. Indeed, similar to copepods, euphausiids, jellyfish and chordates, the SR of haptophytes is predicted to decrease in the tropics. Such differences imply that climate change will reshuffle the distribution and richness of functional types within trophic levels.

These differing responses across trophic levels are not a random result due to the stacking of highly variable species-level patterns. The ten PFGs investigated show distinguishable responses of SR to environmental forcing as their respective top-ranked predictors vary (Supplementary Note 1). Our results are in line with the findings of Ibarbalz et al.[10] who also reported

SST-driven latitudinal diversity gradients for photosynthetic protists and copepods, though their modeled diversity gradients differed less across trophic levels. Our contemporary and future projections of phytoplankton and copepod SR were positively correlated to their estimates of contemporary and future changes in mean annual photosynthetic protists and copepod diversity (see Supplementary Note 6). The predicted responses of global phytoplankton and copepod diversity to future warming are congruent across the two studies, but the amplitude of these predicted changes in diversity differ greatly (Supplementary Note 6). Contrary to our projections, Ibarbalz et al.[10] predicted increases in photosynthetic protists diversity in the North Atlantic and Arctic Ocean, and a higher proportion of copepod diversity increases in the tropics. The vast methodological differences between our approaches could help explain such differences. Our taxonomy-based approach focuses on the dominant species, which actually define biogeographical patterns and drive ecosystem functioning[30]. Meanwhile, their molecular species diversity estimates (i.e., Shannon indices) based on operational taxonomic units are more inclusive of rare taxa and based on a set of two cruises that were latitudinally and seasonally confined (Supplementary Note 6).

The major contribution to the uncertainty (i.e., variance between ensemble members) of our estimates stem from the choice of SDMs, followed by the uncertainty associated with the ESMs (Supplementary Fig. 4). In comparison, the contribution of the different variable pools to uncertainties is small. This ranking is in line with previous results that indicated SDMs and ESMs to be the factors influencing uncertainties the most[29]. Ensemble members often agree on the sign of the SR response, but differ in their projected amplitudes as a function of their sensitivity to the RCP8.5 forcing (Supplementary Fig. 4) and the warming-driven emergence of non-analog environmental conditions (Supplementary Fig. 5). The most sensitive ensemble members predict disproportionally high increases in phytoplankton SR in regions where monthly SST exceeds those prevailing in the contemporary ocean (Supplementary Figs. 4 and 5).

**Temperature is the main driver of changes in species diversity**. Analyses of the predictor dominance in our SDMs reveal that surface temperature is the main driver for both the contemporary distribution of SR and its future changes (Supplementary Note 1). This is in line with previous findings pinpointing temperature as the primary control on marine ectotherm diversity, as well as its changes in a warming ocean[10–13,15]. The link between temperature and SR is consistent with the metabolic theory of ecology (MTE[12,14,31]), which predicts that the natural logarithm of SR scales with the available thermal energy with a slope of ~0.32 (eV)$^{-1}$ for autotrophs and ~0.65 (eV)$^{-1}$ for heterotrophs[31]. For phytoplankton (Fig. 2a) a linear fit to the log of SR for temperatures >22 °C gives a slope of 0.33 (eV)$^{-1}$, and for zooplankton (Fig. 2b) the slope between 11 °C and 20 °C amounts to 0.66 (eV)$^{-1}$. Our results are consistent with previous work on phytoplankton reported in[11] (slope of −0.37 (eV)$^{-1}$ for phytoplankton and temperatures >19 °C).

Drivers of diversity vary across spatio-temporal aggregation scales, which may explain some of the differences in driver rankings between marine studies. A recent model-based study identified nutrients supply rates, mixing and trade-offs in resources requirements as the main drivers of plankton SR[32]. Their simulation results predict a peak of SR at temperate latitudes (50°), which is in contradiction with most global surveys[10–15]. At the scale of our study, we found variables related to water mixing (e.g., wind stress and mixed-layer depth) and resource availability (e.g., nutrients concentrations, surface irradiance, oxygen concentration) to be slightly

weaker predictors of species distributions compared to SST although they still often rank among the top predictors for many functional groups (Supplementary Note 1). Overall, we consider our result of temperature being the key driver as robust at the scale of investigation, which gives us confidence with regard to the use our empirical model for extrapolation into the future.

Since the distribution of the future surface warming is relatively uniform, it is the diagnosed non-linear temperature-diversity relationship (Fig. 2), and in particular the deviations at tropical temperatures from the linear slope predicted by the MTE, that determine the non-uniform response of phytoplankton SR and zooplankton SR to climate change. For zooplankton, its steeper slope leads to larger increases in SR at the thermal range prevailing in temperate to subpolar regions. The inflection point found for zooplankton at ~25 °C translates into a reduction of zooplankton SR in the tropics, as the level of warming in the RCP8.5 scenario pushes many species beyond their thermal tolerances (Fig. 1k). The inverted slope of SR at very low temperatures (Fig. 2) explains the reduction in SR at the very high latitudes.

**Warming will reshuffle community composition**. The most pronounced climate change signal in plankton SR is the strong increase at temperate to subpolar latitudes (~50°N, Fig. 1). By analyzing the species-specific shifts in the centroids of the suitable habitats between now and the future, we find that this increase is primarily driven by the poleward shift of tropical and subtropical plankton species[16–18]. It turns out that 79% of all species shift poleward with these shifts being more common for zooplankton (87%) than for phytoplankton (67%), reflecting a higher sensitivity of zooplankton SR to temperature (Fig. 2). The overall poleward median shift velocity is found to be 35 ± 22 km/decade with no significant difference between phytoplankton (34 ± 28 km/decade) and zooplankton (36 ± 20 km/decade). These velocities are in line with those observed for multiple marine clades and are consistent with the shifts of surface isotherms[16–18].

These poleward range shifts lead to a major restructuring of the plankton community composition, whose amplitude cannot be assessed from changes in SR alone (Fig. 1). Indeed, a community that experiences the replacement of all its constituting species by an equivalent number of newcomers will display a 100% rate of turnover but no change in richness. To investigate the strength of global plankton species turnover triggered by climate change, we examined future changes in species composition using the true turnover (ST, Fig. 3a–f) component of Jaccard's dissimilarity index (see "Methods"). Phyto- (Fig. 3a, b) and zooplankton (Fig. 3c, d) display similar patterns in ST, leading to a global median ST in plankton of 18 (±10)% (Fig. 3e, f). Globally, median ST is slightly higher for zoo- than for phytoplankton (Kruskal–Wallis test; Chi$^2$ = 2577; $p < 10^{-3}$). A sharp latitudinal gradient in ST is predicted as rates are lower in the tropics (16% ± 7.9) than in latitudes >60° (45% ± 16), where SR increases are strongest (Fig. 1). This confirms that climate change triggers substantial species replacement as a consequence of poleward shifts. The majority of the Arctic Ocean plankton community exhibits ensemble mean ST rates >45%, highlighting its high sensitivity to climate warming as the latter is triggering a rapid borealization of the Arctic ecosystems beyond the fish community[33].

The climate-driven ST leads to major changes in species associations, i.e., the potential interactions within the plankton[20,21] (Fig. 3g–j). Using a text analysis algorithm to identify those species pairs that co-occur more frequently than expected given their individual occurrence in the contemporary and future ocean, we find that ~40% of all species associations are reshuffled. Phyto- and zooplankton show opposing gain and loss patterns: changes in SR translate into a 28% gain and 10% loss in

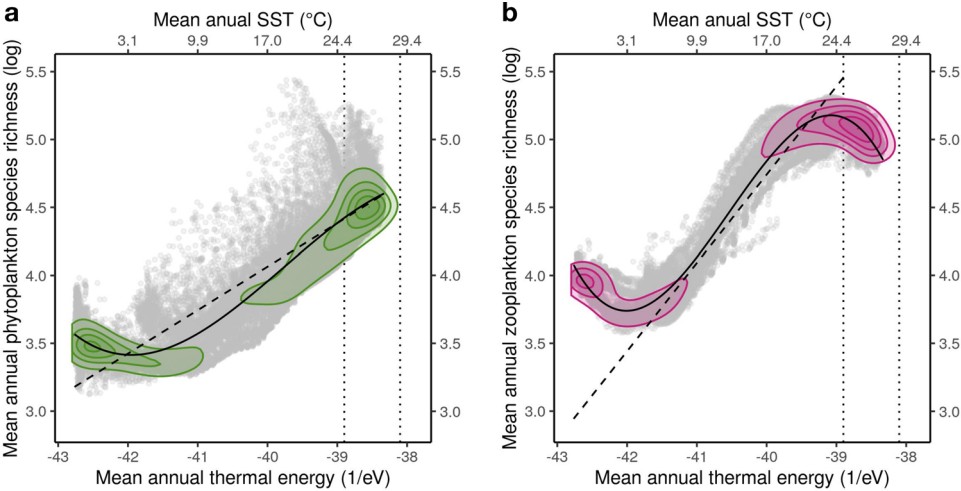

**Fig. 2 Comparing the relationship between species diversity and temperature between phyto- and zooplankton.** Relationships between mean annual phytoplankton (**a**) and zooplankton (**b**) species richness (log-scale) and contemporary surface temperature scaled as available thermal energy according to the framework of the metabolic theory of ecology (MTE). The natural log of species richness (SR) displays a non-linear relationship as a function of contemporary surface thermal energy, which is derived from mean annual sea surface temperature (SST, in Kelvin) and Boltzmann's constant ($k$). The dashed lines illustrate the global linear relationship predicted from the slopes expected from the MTE (~0.32 for phytoplankton; ~0.65 for zooplankton). The solid curves illustrate the 3rd degree polynomial fit that best explains the global variations of log(SR) as a function of mean annual available thermal energy. The colored isopleths illustrate the density of ocean grid cells, based on two dimensional kernel density estimates, and were used to highlight the parts of the gradients driving the observed non-linear relationships. The vertical dotted lines indicate the range of SST prevailing in the tropical band (i.e., latitudes < 30°) for the end-of-century period, according to the ensemble of earth system models forced by a RCP8.5 greenhouse gas emission scenario.

phytoplankton associations, whereas 10% of the zooplankton species associations are projected to be gained and 26% to be lost. Overall, a median of 27% of the future phytoplankton–zooplankton associations represent potentially novel species interactions resulting from the projected spatial shifts and changes in composition.

**Overlap of diversity changes with marine ecosystem services.** Our approach does not allow to quantify explicitly how these potential novel associations will cause changes in the food-web, but the regions with the largest changes in the plankton community are those that provide some of the highest levels of ecosystem services (Fig. 4). To investigate these regions and links, we first defined a severity index by retrieving the two first principal components of a Principal Component Analysis (PCA) that summarize 82.5% of the total variance in changes in phyto- and zooplankton SR and the associated ST rates ("Methods"). Using these principal components, we clustered the global open ocean into six regions and assess ecosystem services provisioning across them (Fig. 4a, b). Six variables linked to marine ecosystem services were considered: the species richness of marine megafauna[13], mean annual catch rates of small (<30 cm) pelagic fishes[34], mean annual net primary production (NPP)[35], the corresponding fraction and efficiency of the production exported below the euphotic zone, and an index of annual plankton size[36] (Fig. 4b). In addition, we derive estimates of microphytoplankton and zooplankton community size structure from species-level measurements and the communities modeled for the contemporary and future ocean. We thereby evaluate how changes in plankton species diversity affect the services variables, as size is a functional trait regulating most ecosystem functions (Supplementary Note 10 and 11).

Changes in mean annual plankton SR, composition and size structure are the most severe in temperate latitudes and in the Arctic Ocean as a consequence of poleward migrations[16–18,33]. The SR of all PFGs is projected to increase in temperate latitudes (Supplementary Note 6 and 9), suggesting that strong increases in SR might promote functional richness. Furthermore, our analysis

highlights that the two most sensitive regions are also key contributors to carbon cycle-related services[8,35] and small pelagic fisheries[34] as their current plankton communities comprise larger organisms[36] that efficiently export organic carbon[6–8] and constitute important food for fishes[2]. For example, we find that smaller warm-water diatoms and copepods species will replace larger ones at high latitudes (Supplementary Note 10 and 11). This likely will weaken carbon export efficiency either because species-poor communities dominated by larger cells/body sizes do not optimize the use of resources, or because larger organisms display functional traits that promote the export of organic matter (Supplementary Note 10 and 11[2,4,6,8]).

Conversely, the Southern Ocean, the Peruvian upwelling and transitional regions emerge as the least sensitive areas, meaning that certain regions with highest carbon export efficiency[4,7,8,35] may remain weakly affected by future changes in plankton diversity. Subtropical gyres, some tropical upwellings and parts of the NE Pacific Ocean display intermediate severity in plankton diversity changes as they exhibit divergent responses of phyto- and zooplankton SR on top of weaker ST. These regions contain valuable hotspots of marine macrofauna biodiversity[13], which shows a strong positive connection to the diversity of all PFGs (Supplementary Note 9). Marine megafauna diversity shows a relation to temperature that is very similar to the one found for zooplankton (Fig. 2[12,13],). Therefore, higher trophic species groups will likely migrate poleward as a result of warming[16–18,33] since their diversity seems to be driven by similar large-scale processes as zooplankton diversity.

Our projected changes in plankton SR as well as the major restructuring of open ocean plankton communities in response to climate change exposes yet another major threat for marine ecosystems associated with the ongoing anthropogenic emissions of greenhouse gases[6,16,18,37]. We find that the biological carbon pump is weakened under higher levels of plankton diversity, which are promoted by higher temperature and resource limitation. Alternatively, the biological carbon pump is more efficient under species-poor communities characterized by higher proportions of larger

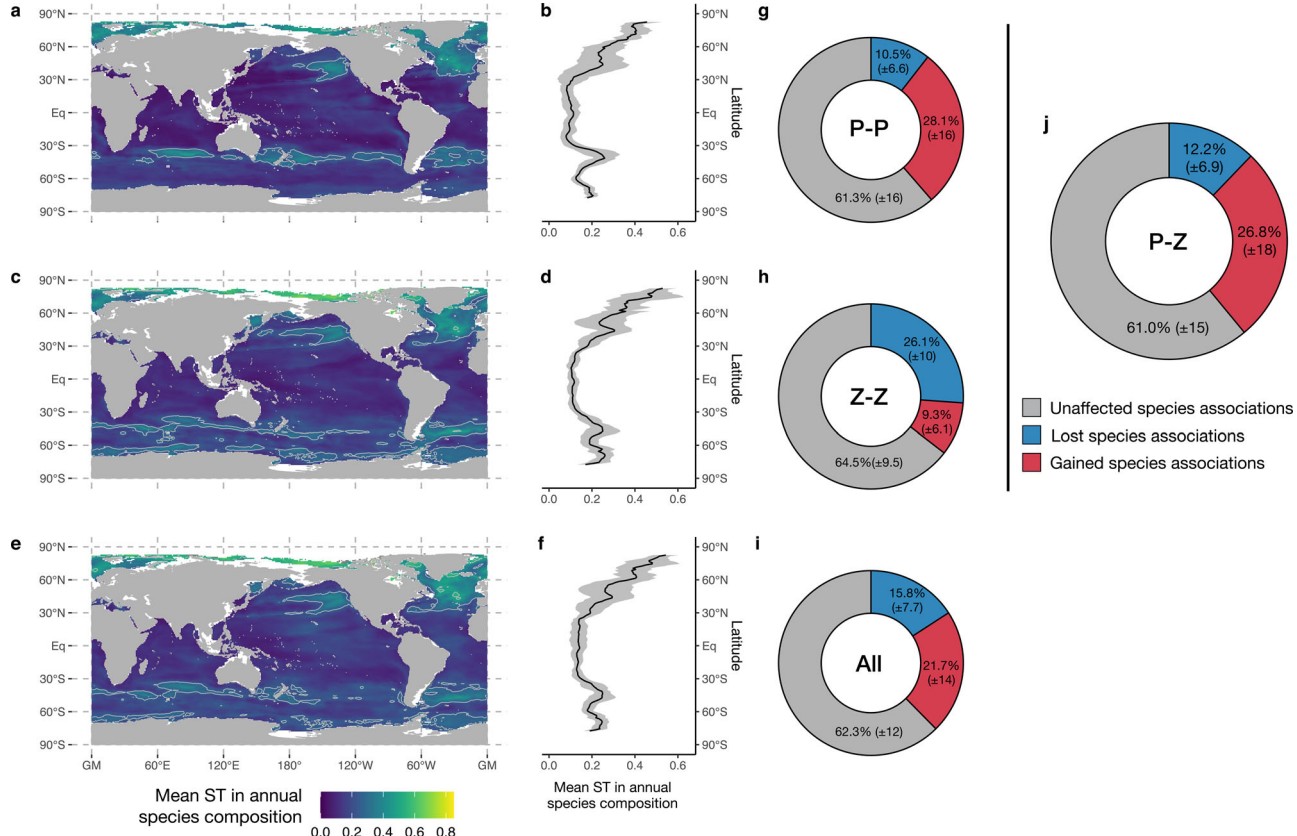

**Fig. 3 Global patterns of species turnover (ST, true turnover component of Jaccard's dissimilarity index) and changes in species associations between 2012–2031 and 2081–2100. a** Map of annual mean ST of phytoplankton species and its zonal average (**b**). **c** and **d** same as **a** and **b** but for zooplankton. **e** and **f** same as **a** and **b** but for all plankton. **g** Changes in phytoplankton species associations with the color indicating the percentage of species associations that are lost (blue), gained (red), or that remain constant (gray). **h** same as **g** but for zooplankton. **i** same as **g** but for all plankton. **j** as **g** but for the associations between phyto- and zooplankton species. The median (±IQR) relative contribution (%) of the three types of species associations to the total number of species associations identified as significant in the contemporary and future ocean are given in **g**, **h**, **i**, and **j**. The pale contour lines in **a**, **c**, and **d** indicate the isopleths of 25% mean ST. The gray contours in **b**, **d**, and **f** illustrate the standard deviation (std) associated with the zonal average displayed by the bold line. Total sample size is $N = 35{,}023$ grid cells.

species. Given that the most severe changes in plankton diversity and size structure occur in regions with elevated levels of plankton-mediated ecosystem services, the risks for disruptive developments, and perhaps even collapses of ecosystem functioning are substantial[38]. Improving the monitoring of marine plankton diversity and the models through which we assess this diversity is imperative.

## Methods

**Overview**. Our study investigates the patterns and drivers of global marine plankton diversity by simultaneously modeling the spatial distribution of 860 phyto- and zooplankton species, based on the widest and most recent compilations of in situ observations available. These observations were associated with various sets of relevant predictors to train a range of statistical species distribution models (SDMs) on a monthly resolution. The SDMs were used to estimate contemporary and future levels of global surface species richness (SR) for total plankton, phytoplankton and zooplankton. We explore how, and why, global phyto- and zooplankton SR and community composition are affected by future climate change under the RCP8.5 scenario of greenhouse gas (GHG) emissions. We also summarize regional patterns of climate change impacts on plankton diversity by clustering the global ocean and examine how hotspots of climate change impacts might overlap with the current provision of marine ecosystem services. All data manipulation and analyses were performed under the R programming language[39]. The R packages used are mentioned below in their corresponding section.

*Plankton species observations*. First, to model global, open ocean plankton diversity from species-level field observations, comparable datasets of phytoplankton and zooplankton occurrences (i.e., presences) had to be compiled. We refer to as open ocean all those regions where the seafloor depth exceeds 200 m. We made use of the large dataset of phytoplankton occurrences recently compiled by Righetti et al.[11]. For zooplankton, a new dataset was compiled following the same methodology. Both

occurrence datasets were based on publicly available data from online biodiversity repositories, as well as some additional published datasets. The R packages mainly used for implementing the datasets are those constituting the tidyverse package.

### Phytoplankton occurrences

For the phytoplankton occurrences used here, Righetti et al.[40] compiled data from various sources: the Global Biodiversity Information Facility (GBIF; https://www.gbif.org/), the Ocean Biogeographic Information System (OBIS; https://www.obis.org), the data from Villar et al.[41], and the MAREDAT initiative[42]. Righetti et al.[40] gathered >10^6 presences from nearly 1300 species sampled through various methodologies within the monthly climatological mixed-layer depth, at an average depth of $5.41 \pm 6.95$ m (mean ± sd), between 1800 and 2015. The species names were corrected and harmonized following the reference list of Algaebase (http://www.algaebase.org/) and were further validated by expert opinion. The final species list spanned most of the extant phytoplankton taxa composing the biodiversity of the euphotic zone of the global ocean. Fossil records, sedimentary records, and occurrences associated with senseless metadata were removed. This dataset has been mined to effectively obtain phytoplankton SR estimates for the global open ocean that were: (i) robust to sampling spatial-temporal biases[11], and (ii) validated against independent data[11].

### Zooplankton occurrences

A new dataset of global zooplankton species occurrences was compiled in a comparable fashion to that put together for phytoplankton. Prior to retrieving the occurrence data online, we first identified the phyla (Order/Class/Family) that comprise the bulk of extant oceanic zooplankton communities: Copelata (i.e., appendicularians), Ctenophora, Cubozoa (i.e., box jellyfish), Euphausiidae (i.e., krill), Foraminifera, Gymnosomata (i.e., sea angels, pteropods), Hydrozoa (i.e. jellyfish), Hyperiidea (i.e., amphipods), Myodocopina (i.e., ostracods), Mysidae (i.e., small pelagic shrimps resembling krill), Neocopepoda, Podonidae and *Penilia avirostris* (i.e., cladocerans), Sagittoidea (i.e., chaetognaths), Scyphozoa (i.e., jellyfish), Thaliacea (i.e., salps, doliolids and pyrosomes),

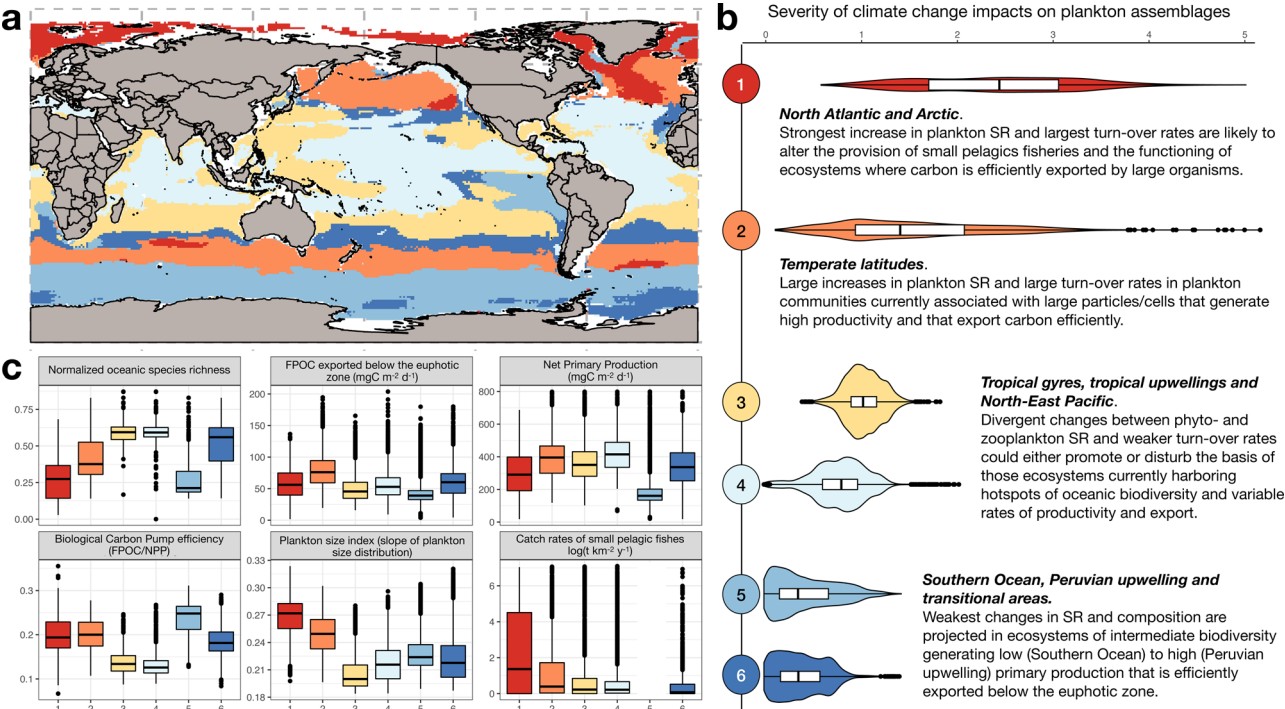

**Fig. 4 Overlap analysis between climate change impacts and marine ecosystem services provision.** Distribution of **a** the ocean regions defined and **b** ranked according to the median severity index of climate change impacts on their plankton community and **c** how they overlap with the contemporary provisioning of marine ecosystem services. The regions were defined by clustering every raster cell of the global ocean based on their average projected difference in annual phyto- and zooplankton species richness, phyto- and zooplankton species true turnover and total plankton turnover. Six proxy variables linked to marine ecosystem services across the six regions were considered: Oceanic megafauna biodiversity (SR)[13] (normalized species richness), mean annual catch rates of small (<30 cm) pelagic fishes (log(tons km$^{-2}$ yr$^{-1}$))[34], annual net primary production (NPP; mgC m$^{-2}$ d$^{-1}$)[35], the corresponding fraction of particulate organic carbon exported below the euphotic zone (FPOC; mgC m$^{-2}$ d$^{-1}$) and the corresponding efficiency of the production exported (FPOC/NPP), and an index of mean annual plankton size[36]. **b** also summarizes how the changes in plankton richness and composition might impact the marine ecosystem services shown in **c**. More details are provided in Supplementary Note 2. The lower, middle, and upper boundaries of the boxplots shown in **b** and **d** correspond to the 25th, 50th, and 75th percentiles. The lower and upper whiskers extend from the hinges to the lowest or largest v grid cells alue no further than 1.5*IQR (interquartile range) from the lower and upper hinges. $N = 35,023$ grid cells for the total sample size ($N_{region1} = 2344$; $N_{region2} = 5954$; $N_{region3} = 6748$; $N_{region4} = 8290$; $N_{region5} = 7637$; $N_{region6} = 4050$).

Thecosomata (i.e., sea snails, pteropods), and four families of pelagic Polychaeta (i.e., worms) that are often found in the zooplankton and whose species are known to display holoplanktonic lifecycles (Tomopteridae, Alciopidae, Lopadorrhynchidae, Typhloscolecidae). The presence data associated with species belonging to these groups were retrieved from OBIS and GBIF between the 12/04/2018 and the 18/04/2018 using online queries via the R packages RPostgreSQL, robis and rgbif. Since the Neocopepoda infraclass comprise several thousands of benthic and parasitic taxa[43], a preliminary selection of the non-parasitic planktonic species had to be carried out prior to the downloading using the species list of Razouls et al.[43] as a reference. The spatial distributions of the groups cited above were first inspected using GBIF's and OBIS's online mapping tools to evaluate the potential number of overlapping observations between the two databases. As a result of their relatively low contributions to total observations/diversity, and very high overlap between databases, the occurrences of Cladocera and Polychaeta were retrieved from OBIS only (which usually harbors more occurrences). On top of the data collected from OBIS and GBIF, the copepod occurrences from Cornils et al.[44] and the pteropod occurrences from the MAREDAT initiative[45] were added to the dataset. This initial collection of zooplankton observations gathered 4,899,151 occurrences worldwide. Then, similar criteria as Righetti et al.[11,40] were applied to progressively remove those presences that would be discordant with estimates of contemporary open ocean zooplankton diversity. The number of observations and species discarded after each main step and for each initial dataset are reported in Supplementary Data 1. We discarded records that: (i) presented at least one missing spatial coordinate, (ii) were associated with an incomplete sampling date (d/m/y), (iii) were associated with a year of collection older than 1800, (iv) were not associated with any sampling depth, (v) were not identified down to the species level, and (vi) were issued from drilling holes or sediment core data. For step (vi), a list of keywords (Supplementary Note 3) was used to identify the names of those original datasets that contained either fossil or sedimentary records. These first steps resulted in the removal of 1,766,783 occurrences (~36%). Like for phytoplankton, the remaining occurrences were associated with surface salinity values from the World Ocean Atlas (WOA) 2013[46] and bathymetry levels from the National Oceanic and

Atmospheric Administration (NOAA) using the marmap R package. Occurrences associated with salinity levels <20 and seas shallower than 200 m were removed to only keep presences within the open ocean. These two steps removed 1,435,108 occurrences (~46%). To restrict observations to those occurrences collected in the environmental conditions prevailing in the euphotic zone, or the mixed layer, we discarded occurrences sampled with a net tow whose maximal sampling depth was >500 m. The average depth was used when maximal depth was not provided in the metadata. Therefore, the maximal depth of a zooplankton species occurrence allowed in our dataset is 500 m. This way, we tried to account for the zooplankton community that frequently performs diel vertical migration across the euphotic zone or the mixed layer, and that often co-occurs with species inhabiting surface layers. This removed 109,582 (~6%) occurrences. Next, for each phylum, OBIS and GBIF datasets were merged and the list of species names were extracted. Every species name was then carefully examined and compared to the taxonomic reference list of the World Register of Marine Species (WoRMS; http://www.marinespecies.org) for all taxa but copepods, for which we used the taxonomic reference of Razouls et al.[43]. This way, we rigorously harmonized and corrected the species names across all datasets. In addition, we used the notes and attributes of WoRMS to identify whether species were holoplanktonic or meroplanktonic (i.e., those species that have at least one benthic phase in their life cycle). Jellyfish species usually display a fixed polyp phase during their life cycle, therefore we used the dataset of Gibbons et al.[47] to remove the species that were not holoplanktonic. Overall, these steps discarded 37,234 occurrences (~2%). One of two duplicate occurrences were removed from the dataset if they displayed the same species name, sampling depth, sampling date, and if they occurred within the same 0.25° x 0.25° cell grid. This last step removed 900,446 occurrences (~54%), highlighting the high overlap between the two main data sources. The remaining 766,033 presences were binned into the monthly 1° x 1° grid cell of the WOA to match the spatial resolution of the environmental predictors. The average maximum (±std) sampling depth was 73 ± 109 m and the average sampling year was 1985 ± 21. Observation densities were spatially biased towards the North Atlantic Ocean and the Southern Ocean (Supplementary Figure 1). The data reflected the historical

seasonal sampling bias towards spring and summer. In the northern hemisphere, observations were equally distributed from March to October but constituted 78% of the data. In the southern hemisphere, 75% of occurrences were sampled between November and March. The final dataset gathered occurrences for 2034 different species (576 genera, 161 families) spanning all the major zooplankton phyla and several size classes. The only notable missing taxa are those belonging to the Cercozoa and Radiozoa because they present little to no species-level observations in online biodiversity repositories as they have been historically overlooked by traditional sampling techniques[48].

*Contemporary environmental conditions.* A comprehensive set of environmental variables that are known to affect the physiology and constrain the distribution of plankton was prepared to define the candidate predictors for the SDMs. The R packages mainly used were raster and ncdf4. First, twelve primary variables that are relevant for modeling the distribution of both phytoplankton and zooplankton taxa were identified[49–52]. These were then aggregated into twelve monthly climatologies at a 1° x 1° resolution (i.e., the spatial cell grid of the WOA). The first six primary variables were sea surface temperature (SST, °C), sea surface salinity (SSS), nitrates ($NO_3^-$), phosphates ($PO_4^{3-}$) and silicic acid ($Si(OH)_4$) surface concentrations (µM), as well as dissolved oxygen concentration ($dO_2$, ml l$^{-1}$). Oxygen limitations and oxygen minimum zones (OMZ) are key factors controlling the horizontal and vertical distribution of zooplankton[53,54]. However, the effects of oxygen are often confounded with those of temperature because surface oxygen scales linearly with SST on a global scale. Therefore, $dO_2$ at 175 m depth was used instead of surface $dO_2$. For all six variables, the twelve monthly climatologies of the WOA13v2 (https://www.nodc.noaa.gov/OC5/woa13/woa13data.html) were used. In addition, satellite observations stemming from the Sea-viewing Wide Field-of-view Sensor (SeaWiFS; https://oceancolor.gsfc.nasa.gov/) over the 1997 to 2007 time period were used to derive monthly climatologies of Photosynthetically Active Radiation (PAR; µmol m$^{-2}$ s$^{-1}$) and chlorophyll (Chl; mg m$^{-3}$), the latter serving as a proxy for surface phytoplankton biomass. Monthly climatologies of mixed-layer depth (MLD, m) based on the temperature criterion of[55] from the Argo floats data (http://mixedlayer.ucsd.edu/) were also considered. Climatologies of surface wind stress (m s$^{-1}$) were obtained from the Cross-Calibrated Multi-Platform[56], using data from 1987 to 2011 (https://podaac.jpl.nasa.gov/). Climatologies of surface carbon dioxide partial pressure (pCO2; atm) were obtained from the Surface Ocean CO$_2$ Atlas (SOCATv2; https://www.socat.info/) and made available by Landschützer et al.[57]. Lastly, a variable depicting sub-mesoscale dynamics and the strength of sea currents was derived from the daily satellite altimetry observations over the 1993–2012 period (https://cds.climate.copernicus.eu/#!/home): mean Eddy Kinetic Energy (EKE, m$^2$ s$^{-2}$). EKE was computed from the northward and eastward components of surface geostrophic seawater velocity (assuming the sea level as geoid), following the method of Qiu & Chen[58]. Such variable enabled us to account for the potentially important role of sub-mesoscale activity in structuring plankton biodiversity[59].

Then, nine secondary predictors were derived from some of the predictors described above. PAR over the MLD (MLPAR, µmol m$^{-2}$ s$^{-1}$) was calculated following Brun et al.[49] An estimate of annual range of SST (dSST) was added by computing the difference between the warmest and the coldest temperature across the 12 months. The excess of nitrate to phosphate (N$^*$, µM) relative to the Redfield ratio was computed as $[NO_3^-] - 16[PO_4^{3-}]$. Changes in N$^*$ represent varying conditions of denitrification and remineralization from N$_2$-fixing organisms[7]. The excess of silicates to nitrates (Si$^*$=$[Si(OH)_4]$-$[NO_3^-]$, µM) was also computed to represent regions where silicates are in excess compared to what diatoms would need to use up the nitrates[7]. Si$^*$ > 0 are indicative of conditions where diatoms can grow healthy. Since the distribution of macronutrients concentrations, chlorophyll concentration, and EKE values were all skewed towards lower values, we considered their logarithmic values (logNO3, logPO4, logSiOH4, logEKE, and logChl), based on either natural log or base 10, as additional predictors because they were much closer to a normal distribution.

*Species distribution modeling.* SDMs refer to a wide range of statistical algorithms that link an observed biological response variable (i.e., presence-only, presence/absence, abundance) to contextual environmental variables in the form of a response curve[60]. The latter is used to explore how a species' environmental niche is realized in space and time[24]. In short, SDMs mainly rely on the following assumptions: (i) species distributions are not strongly limited by dispersal at a macroecological scale, an assumption valid for plankton considering the very strong connectivity of ocean basins through surface current on decadal scales[61,62], which enables plankton species to display very large spatial ranges;[11,47,60] (ii) species distributions are primarily shaped by the combinations of environmental factors that define the conditions allowing a species to develop. The latter assumption has been supported on macroecological scales, where the imprint of biological interactions (and dispersal) has been found to be relatively small[63,64]. Neither comparable abundance data nor presence/absence data were available from our datasets. In addition, presence-only data are less sensitive to discrepancies in species detection across various plankton sampling techniques. Therefore, based on species presences, we developed an exhaustive SDM framework to estimate plankton diversity patterns from an ensemble modeling approach[65] that addresses the underlying main sources of uncertainties[66,67].

We follow the methodology developed in ref. [11], but simplify this approach to accommodate the limited predictor availability in the future model projections (see

section "Choice of environmental predictors"), and the large number of diverse species we model in this work (sections "Background data (pseudo-absences)", "Choice of environmental predictors", and "SDMs evaluation and projections of monthly plankton species community composition"). We further derive SR based on habitat suitability rather than observed presence–absence data (section "Ensemble projection of global plankton species richness"). All methodological choices led to a minimization of computational cost and model complexity, while preserving all crucial patterns reported in ref. [11]. Each methodological choice was carefully evaluated against other options, see sections below.

## Background data (pseudo-absences)
Since we aimed at training correlative SDMs to model species distributions from presence data and environmental predictors, background data had to be simulated to indicate those conditions where a species is likely not to occur (i.e., pseudo-absences[68]). The generation of background data is a critical step in niche modeling experiments, and though no single optimal method has been identified by the niche modeling community, this step must address the important spatial and temporal sampling biases inherent to field-based observations. To do so, we made use of the target-group approach of Philipps et al.[69], which has been shown to efficiently model phytoplankton distributions[11]. This method was found appropriate for our study because it generates background data according to the density distribution of the presence data, and therefore it: (i) does not induce additional bias to the initial biases in the presence data; and (ii) does not misclassify regions lacking observations (e.g., South Atlantic and Subtropical Pacific, Supplementary Fig. 1) as regions of absences.

For phytoplankton, we followed the background selection procedure described in Righetti et al.[11]. The authors used either the total pool of occurrences as a target-group, or defined three target groups based on the taxonomic groups contributing most to species diversity and observations (Bacillariophyta, Dinoflagellata, and Haptophyta). Background data of each species were randomly drawn based on the monthly resolved 1° x 1° occurrences of both their corresponding target groups, after applying an environmental stratification based on the SST and MLD gradients. This way, a species' background is located at the sites where its lack of presence is most likely to reflect an actual absence. For each species, ten times more background data than presences were generated following the guidelines of Barbet-Massin et al.[68]. The amount of background data sampled from a specific SSTxMLD stratum was proportional to the number of monthly 1° x 1° cells provided by the target-group in this very stratum, thereby reflecting original sampling efforts. Both the total target-group background data (drawn from all sampling sites together) and the group-specific target background data were considered for our study, but both led to comparable estimates of phytoplankton species diversity (but see Fig. S3B of ref. [11]).

The same method was applied for zooplankton species. First, we defined target groups based on their sampling distribution and broad taxonomic classification: Arthropoda (mainly copepods, but also krill and amphipods, that are sampled through similar techniques), Pteropoda, Chaetognatha, Cnidaria, Ctenophora, Chordata, Foraminifera, and Annelida. Unfortunately, the last three target groups displayed too few occurrences for drawing ten times more background data than presences. Consequently, their background data were drawn from the total pool of occurrences. Ctenophora also showed very few observations so they were merged with the Cnidaria as they are often considered together as jellyfish and collected in similar ways.

Total and target-group background data were drawn for all zooplankton species presenting more than 100 presences to run preliminary SDMs based on a preliminary set of predictors (Supplementary Fig. 2). These SDMs were then used to project preliminary diversity patterns for the four months that represent each season in the northern hemisphere (April, July, October, and January). These projections were examined for every group, and the predictive skills of the SDMs were evaluated using a repeated ten times split-sample test (see below). These tests showed that the total target-group background and the group target-group background converged towards SDMs of comparable skills and similar diversity patterns, except for the Chaetognatha, for which the target-group background leads to models of much poorer predictive skills (Supplementary Fig. 2). Furthermore, both background choices led to very similar latitudinal diversity gradients for phytoplankton (Fig. S3B in ref. [11]). Therefore, to generate diversity patterns that are robust and consistent across the two trophic levels, the total target background data were used as standard background. The phytoplankton diversity pattern obtained with the total target background approach was only slightly lower in the Indo-Pacific and at very high latitudes[11]. Once they were generated, all background data were matched with monthly values for the 21 environmental variables described above.

## Choice of the SDMs algorithms
The choice of the statistical method is a main source of uncertainty when projecting biodiversity scenarios through niche modeling[66,67]. Therefore, an ensemble forecasting strategy was adopted based on four types of SDMs that cover the range of algorithms types and model complexity that are commonly used:[67,69] Generalized Linear Models (GLM), Generalized Additive Models (GAM), Random Forest (RF), and Artificial Neural Networks (ANN). The level of complexity of those models was constrained to avoid model overfitting[70], a common pitfall when dealing with noisy and spatially biased data. SDMs including numerous predictors and parameterization features are more likely to fit

spurious relationships and to be less transferable[70,71]. Consequently, the number of predictors was limited relative to the number of presences (see below) and the SDMs were tuned to fit relatively simple response curves. The GLM followed a binomial logit link, including linear and quadratic terms, and a stepwise bi-directional predictor selection procedure. The GAM also followed a binomial logit link. Smoothing terms with five dimensions, estimated by penalized regression splines without penalization to zero for single variables, were applied. Interaction levels between environmental predictors were set to zero for both GLM and GAM. The RF included 750 trees, and terminal node size was fixed at 10 to avoid having single occurrences as end members of some trees. The number of variables randomly sampled as candidates at each tree split (mtry parameter) was equal to the number of predictors used divided by three. The numbers of units in the hidden layers of the ANN, as well as the decay parameter, were optimized through five different cross-validations and a maximum of 200 iterations. Background data were weighted inverse-proportional to that of presence data (total weight = 1).

## Choice of environmental predictors

To select for parsimonious and ecologically relevant sets of environmental predictors, a three-stage hierarchical selection framework was developed: (i) the distribution of the predictors' values fitted to the presences were compared to their realized distribution between the main ocean basins to check whether one predictor could bias SDMs outputs towards a particular basin; (ii) pair-wise rank correlations between variables were examined, and one of two collinear variables was discarded where necessary; and (iii) models were trained to evaluate the explanatory power of several predictors sets of increasing parsimony, and rank the predictors within those sets at species-level. This selection procedure was carried out by separating phytoplankton from zooplankton since: (i) the two groups show different sampling distributions, and (ii) their niche dimensions might differ because of differences in their lifecycles (few days for phytoplankton, months to years for zooplankton) and biological requirements (photo-autotrophy vs. heterotrophy and respiration). For those tests, only well-observed species with >100 occurrences were selected ($n_{phytoplankton} = 328$; $n_{zooplankton} = 372$). Ultimately, to account for the uncertainty in predictors choice, several final sets of predictors were defined based on the steps of the selection framework, and ensemble forecasting was adopted again (i.e., diversity estimates will be averaged across the sets of predictors).
*Removal of variables impacted by sampling imbalances across ocean basins:* Imbalance of sampling effort in geographical space can lead to sampling imbalance in environmental space if portions of an environmental gradient are strongly connected to an ocean basin that has been surveyed more extensively than others. To avoid such issues, the distributions of the annual values of the predictors were examined between the main basins (Arctic, Southern, Pacific, Indian and Atlantic Oceans). The most spatially imbalanced predictors were SSS and $pCO_2$: the former is on average higher in the Atlantic Ocean, while the latter exhibits many of its most extreme values in the Peruvian upwelling system (Supplementary Note 3). The Peruvian upwelling is a hotspot of phytoplankton observations with clearly skewed observations (and the number of species sampled) towards $pCO_2$ values > 400 atm (Supplementary Note 3). Plus, the $pCO_2$ data do not cover the Arctic Ocean, the Mediterranean Sea, and the Red Sea. Consequently, $pCO_2$ was discarded from the list of predictors to avoid strong sampling bias effects on SDM projections. A majority of the zooplankton data are concentrated in the Atlantic Ocean (Supplementary Fig. 1). As a result, the distribution of SSS values fitted to zooplankton occurrences is skewed towards SSS values >35 (Supplementary Note 3). As SSS is commonly used as a predictor for modeling the distribution of zooplankton[47–49], we wanted to further examine its potential to act as a basin indicator rather than a predictor meant to represent an actual environmental control on species distribution. To do so, we performed ensemble SDMs projections for the zooplankton species, based on three variables sets: (i) without SSS, (ii) with SSS, and (iii) with Longitude (0°–360°) instead of SSS. Variables sets (ii) and (iii) led to very similar global zooplankton SR patterns with hotspots in the Atlantic Ocean. On the contrary, (i) led to more balanced zooplankton SR between basins without significantly lowering SDMs skills (Supplementary Note 3). We interpreted this as a bias in environmental space towards the conditions prevailing in the Atlantic Ocean, therefore we chose to discard SSS from the list of predictors.
*Removal of collinear variables:* Strong correlations among predictors can mislead the ranking of variable importance in SDMs[72], so it has become common practice to exclude one of two variables that are highly collinear. Pair-wise Spearman's rank correlation coefficients (ρ) were computed based on the predictors' values fitted to the presences. When two variables exhibited a |ρ| > 0.70, the one displaying the distribution closest to a normal distribution was kept. From phytoplankton occurrences, we identified two clusters of strongly correlated variables: one comprising MLD, PAR, MLPAR (by construction), and wind stress (but PAR and MLD were only correlated at ρ = −0.66); and the other one comprising [NO₃⁻], [PO₄³⁻] and their logged versions. Similar clusters were found from zooplankton data, except that PAR was slightly less correlated to Wind stress (ρ = −0.66) and MLD (ρ = −0.58), and that [NO₃⁻], [PO₄³⁻], plus their logged versions, showed stronger correlations with SST (ρ = −0.80). As [NO₃⁻] is a key factor for structuring planktonic systems[7], and because we aimed to keep the variables sets as consistent as possible between species, logNO3 was kept as a candidate predictor. The variables retained for phytoplankton were: SST, dSST, logEKE, Si*, N*, logSiOH4, logNO3, logChl, wind stress, PAR, MLPAR, and MLD. The last four variables were kept to explore the outcome from alternative choices in the variables sets (but see below). The variables retained for zooplankton were the same but with the addition of $dO_2$.

*Examination of the explanatory power of predictors sets and ranking of predictors:* To further evaluate which subset of these variable subsets are key to model species distributions, GLM and RF were performed for each species for several sets of decreasing complexity (from ten to five predictors), and the adjusted $R^2$ of the models, as well as the ranking of predictors within each set, was extracted (Supplementary Note 1). GLM and RF were used here because they are part of the SDMs that will be used for projections afterwards and because they represent maximally different model complexities among the SDM types used[73]. For GLM, predictor importance was determined according to their absolute t-statistic using the caret R package. For RF, predictor importance was based on the Gini index, which measures the mean decrease in node impurity by summing over the number of splits (across all trees) that includes a variable, proportionally to the number of samples it splits. The ranger R package was used for assessing variable importance with RF models. To keep the variables ranks comparable across predictors sets, rank values were normalized to their maximum. For each model type, the distributions of the models' $R^2$ and the distribution of the predictors' ranks were examined for phytoplankton and zooplankton separately. The same was done between the main groups constituting the phytoplankton (Bacillariophyta, Dinoflagellata and Haptophyta) and the zooplankton (Copepoda, Chaetognatha, Pteropoda, Malacostraca, Jellyfish, Chordata and Foraminifera). This allowed us to identify the most important predictors for modeling the species distributions and to evaluate if a decrease in the models' skill was linked to the removal of certain variables. Group patterns allowed us to test whether different groups differed in their main environmental drivers.
For phytoplankton species, 14 sets of variables were examined (Supplementary Note 1). The first nine aimed to test: (i) the impact of alternative choices between variables that were identified as collinear (wind vs. MLPAR vs. MLD + PAR); (ii) the impact of progressively discarding variables that initially presented lower ranks (logEKE, Si*), and (iii) the impact of choosing logNO3 over logSiOH4, two variables representing global macronutrients availability and that present relatively high correlation coefficient (ρ = 0.59). The last five sets of predictors (10–14) aimed to test the impact of alternatively removing those variables that presented relatively high ranks in the previous sets: SST, dSST, N*, logSiOH4, logChl, PAR. In a similar fashion, 15 sets of variables were tested for zooplankton (Supplementary Note 1). The first ten aimed to test: (i) the impact of choosing wind stress over MLPAR or over MLD + PAR; (ii) the impact of selecting PAR over MLD; (iii) the impact of discarding Si*, N*, logEKE; and (iv) the impact of choosing logNO3 over logSiOH4 (ρ = 0.64). The last five sets of predictors (11–15) aimed to test the impact of alternatively discarding the top five predictors: SST, dSST, dO2, logSiOH4, and logChl.
GLM and RF converged towards similar median variable rankings and evidenced high inter-species variability (Supplementary Note 1). For total phytoplankton, GLM identified the following median ranking across all species: SST > N* >logChl > logSiOH4 and dSST > logNO3 > logEKE > Si* > the PAR/MLD/MLPAR/wind stress cluster. RF ranked predictors in the following median order: SST and N* >logChl > dSST > logSiOH4 > logEKE and logNO3 > Si* > the PAR/MLD/MLPAR/wind stress cluster. Yet, both GLM and RF also identified PAR as a major predictor for Haptophyta, which does not appear in the rankings for total phytoplankton because Haptophyta represented ~9% of species composition only. Since adding PAR does not alter the models' $R^2$ for the Bacillariophyta and Dinoflagellata, it was retained for the final predictors sets. For total zooplankton, GLM ranked predictors in the following median order across all species: SST > dSST and logSiOH4 > logChl and logEKE > dO2 and logNO3 > N* >Si* > PAR and MLPAR > wind stress > PAR and MLD. RF identified the following median ranks: SST > dSST and dO2 > logNO3 > logSiOH4 > logChl and logEKE > N* and Si* > PAR > wind stress > MLD and MLPAR. These median rankings reflected those of the Copepoda since they represented >70% of all zooplankton species. Again, rankings displayed high variance, reflecting high inter-species variability. Overall, based on all the results shown above, eight different final predictors sets were kept for modeling the distribution of phytoplankton (n = 4) and zooplankton (n = 4). In contrast to ref. [11], predictor ensembles were defined across all species rather than for each species. This was due to multiple reasons: (i) predictor availability for future model projections was limited and did not allow for species-specific variable choices, (ii) computational constraints with regard to the total number of ensemble members that could be projected, (iii) the five sets already contain those predictors that explain a majority of the variability in most models, (iv) recent findings from Righetti et al. (in prep.) that the uncertainty due to predictor choice is low for models with optimized background selection.
Phytoplankton:

1. SST, dSST, logChl, N*, PAR, and logNO3
2. SST, dSST, logChl, N*, PAR, and logSiOH4
3. SST, dSST, logChl, N*, PAR, logNO3 and Si*
4. SST, dSST, logChl, PAR, and logNO3

Zooplankton:

1. SST, dSST, dO₂, logChl, and logNO3
2. SST, dSST, dO₂, logChl, and logSiOH4
3. SST, dSST, dO₂, logChl, logSiOH4, and N*
4. SST, dSST, dO₂, logChl, logNO3, and Si*

## SDMs evaluation and projections of monthly plankton species community composition

Only species with more than 75 presences were considered for modeling plankton species distributions ($n_{phytoplankton} = 348$; $n_{zooplankton} = 541$) because we aimed to achieve a

relatively high presence-to-predictors ratio (~15, which is the ratio achieved for a species with 75 presences and five to six predictors) to be more conservative than Righetti et al.[11] (i.e., minimum 24 presences) since we aimed to project the SDMs in future conditions based on a pool of species for which we have high confidence. This is in line with Guisan et al.[60] who suggest to maintain at least a ratio of ten. For each species, each SDM, and each set of predictors, presences and background data were randomly split into a training set (80%) and a testing set (20%) and these evaluation tests were repeated ten times. Therefore, 160 (four SDM types x four predictor sets x ten separate evaluation runs) models were trained per species, resulting in a total of 142,240 SDMs. Model skill was evaluated based on two widely used metrics: the True Skills Statistic (TSS[74]) and the Area Under the Curve (AUC[60]). TSS values range between −1 and 1, with null values indicating that models perform no better than at random. AUC ranges between 0 and 1, with values <0.5 indicating worse than random model skill. To remain somehow conservative and increase our confidence in SDMs projections, only species displaying average TSS values >0.30 were retained for the final ensemble projections (Supplementary Fig. 3). In total, 860 species were considered as successfully modeled ($n_{phytoplankton} = 336$; $n_{zooplankton} = 524$; Supplementary Data 2). For those, each of the 160 SDMs was projected onto the twelve monthly climatologies of its corresponding predictors set and the projections were averaged over the ten cross-evaluation runs. This way, we obtained global maps of monthly mean presence probability for each of the 16 SDM x predictor set combinations. These maps are to be interpreted as habitat suitability patterns that highlight the regions of the global ocean where the environmental conditions are most favorable for a species to develop. Habitat suitability maps were not converted to binary presence–absence maps as probabilistic outputs provide more gradual responses that should better reflect the very dynamic occupancy patterns of plankton and that are better suited than threshold approaches for our purposes[75–77]. For each grid cell of the global ocean, the probabilistic estimates of species habitat suitability were stacked to obtain monthly estimates of species composition. All SDMs were trained, evaluated, and projected using the biomod2 R package.

## Ensemble projection of global plankton species richness

For every SDM x predictor set combination and every month, we summed the species habitat suitability to estimate monthly SR. Then, annual average SR was estimated for each cell. The annual average was preferred over the annual integral because of the high latitudes that presented a lot of missing values in winter because of the lower coverage of satellite products. Since this diversity estimate is the sum of habitat suitability indices, it is to be interpreted as the amount of SR that the monthly/annual average environmental conditions should be able to sustain (i.e., potential SR). This way we obtained 16 estimates of annual SR for total plankton (all 860 species together), phytoplankton and zooplankton. Ensemble projections of annual SR were then obtained for these three categories by averaging the annual SR estimates.

As the biological data used to train the SDMs span several decades (mostly between the 1970s and 2000s), our diversity estimates are integrative of changes in SR and species composition (i.e., changes in beta diversity) that occurred during these decades. The phytoplankton species modeled are mainly members of the Bacillariophyceae (45.8%), and the Dinoflagellata (45.8%), which usually rank among the large marine microalgae. Therefore, the phytoplankton SR estimates shown here should be mainly representative of the microphytoplankton (20–200 µm) rather than smaller size fractions. Nearly half (51.9%) of the zooplankton species modeled are Copepoda, making it the most represented groups in the zooplankton SR patterns followed by: Malacostraca (crustacean macrozooplankton such as Euphausiids and Amphipods; 13.9%), Jellyfish (13.1%), Foraminifera (5%), Chaetognatha (5%), Pteropods (4%), Chordata (3%). The last 4% of species modeled are a mix of Annelids and Branchiopods. The full list of the species modeled as well as their taxonomic classification is given in the Supplementary Data 2. We underline that the 860 species modeled are those for which we have enough records for training reliable SDMs. Therefore, these species are likely to be those that are the most frequently detected by conventional sampling and identification techniques, either because: (i) they are the ones dominating total plankton abundance, which makes their collection more likely; or (ii) they are larger species (in terms of cell volume or body length), which would facilitate their sampling and identification under the binocular or recent imaging systems. We acknowledge that our approach does not allow us to account for rare taxa and thus under samples the true diversity of the marine plankton. Nonetheless, we argue that our approach does allow us to estimate global plankton diversity patterns as the species dominating plankton abundances are those carrying biogeographical information[30], meaning their distribution and abundance patterns can be correlated to environmental gradients. Meanwhile, the patterns of rare and non-dominant species, which constitute the majority local SR, exhibit no biogeographical signature[30]. This has been supported in the previous study of Righetti et al.[11] where the authors showed that global SR patterns were robust to the progressive exclusion of taxa with relatively few records.

We also acknowledge that our estimates of species distribution might be biased by imbalances in species detection and sampling effort between sampling cruises, as those rely on a wide range of collection and identification methodologies. We argue that such biases are particularly significant when relying on abundance data, and that we mitigate them by: (i) converting all observations to presences and aggregating them onto a 1° x 1° grid; (ii) modeling SR as an emergent property overlapping the distribution of single species with equal weighting rather than modeling SR directly, in which case diversity estimates would be highly sensitive to sampling effort imbalances; (iii) by designing the SDMs in a

way that accounts for spatial and temporal sampling biases in geographical and environmental space; and (iv) by tuning down the complexity of the SDMs (i.e., reduced number of features and predictors) in order to avoid model overfitting[70].

*Future environmental conditions in the global surface ocean.* The future monthly fields of the selected environmental predictors were obtained from the projections for the 2012–2100 period of five ESM simulations for the IPCC's RCP8.5 scenario from the MARine Ecosystem Model Intercomparison Project (MAREMIP, http://pft.ees.hokudai.ac.jp/maremip/index.shtml[78]) and/or the Coupled Model Intercomparison Project 5 (CMIP5[79]). The model ensemble contained the following five ESMs (with their embedded ocean and ecosystem models indicated indicated in brackets after the semicolons): Community Earth System Model version 1 (CESM1, POP-BEC), Geophysical Fluid Dynamics Laboratory Earth System Model with Modular Ocean Model version 4 (GFDL-ESM2M; MOM-TOPAZ), Institut Pierre Simon Laplace Climate Model version 5A-LR (IPSL-CM5A-LR; NEMO-PISCES), Centre National de Recherches Météorologiques Climate Model version 5 (CNRM-CM5; NEMO-PISCES) and the Model for Interdisciplinary Research on Climate version 5 (MIROC5; MRI.COM-MEM). All ESMs were fully-coupled except for MIROC5 for which the ocean model was forced by the atmospheric component. All of the projections used here were benchmarked, quality-controlled and described in the previous multi-model comparison studies of Laufkötter et al.[26,80]. Considering the scope of the present study, we refer to these authors' previous extensive descriptions for the full detail of the ESMs used here. Taken together, the present five ESM ensemble gathers models of various sensitivity to future climate forcing, and thus provides a wide range of alternative environmental conditions projected for the future surface ocean. With the present ESM ensemble, we account for the variability in the choice of the climate model, which is known to be a significant source of uncertainty in biodiversity projections; this source being consistently lower than those associated with SDM choice, though[28,66,67].

The monthly projections of the five selected ESMs were interpolated on the 1° x 1° cell grid of the WOA (i.e., the one used to train our SDMs) over the 2012–2100 period for all the nine chosen environmental predictors. To obtain future monthly climatologies that span a comparable amount of temporal variability as the in situ climatologies used to train the SDMs (~20 years), a baseline and an end-of-century time periods were first defined (2012–2031 and 2081–2100, respectively) for every ESM projection run. The 12 monthly climatologies were derived based on the models' monthly projections and monthly anomalies were computed by subtracting the baseline values to the end-of-century ones. For dSST (i.e. annual range of SST), the annual maximum of SST was derived from the monthly climatologies and the difference between the baseline and the end-of-century dSST provided the delta value. These anomalies can be either positive or negative and they represent the difference in the predictors' condition due to future climate change under the RCP8.5 GHG concentration scenario[25]. To obtain the final conditions prevailing in the surface ocean for the end-of-century period, the delta values were simply added to the in situ climatologies representing the conditions in the contemporary ocean. The SDMs of the 860 plankton species successfully modeled were then projected onto these future monthly climatologies for each of the ESM. This way, we estimate the monthly probability-based species composition in the future global ocean for each of the 80 combinations of SDMs ($n = 4$), ESMs ($n = 5$), and predictor set ($n = 4$). Overall, our ensemble forecast approach[65] generates an unprecedented set of 825,600 species-level estimates of global future habitat suitability patterns. Finally, mean annual SR and community composition were calculated for total plankton, phytoplankton and zooplankton for each of the 80 possible combinations of projections, as described in section "Ensemble projection of global plankton species richness".

*Analyses*

## Ensemble projections of changes in species richness, community composition turnover, and changes in species associations between the contemporary and the future ocean

For each of the 80 projection combinations described above, the mean annual SR estimates for the contemporary ocean were subtracted to their corresponding mean annual SR estimates for the future ocean to compute the percentage difference in mean annual SR (%ΔSR) for total plankton, phyto- and zooplankton. The %ΔSR represents the emergent change in SR caused by future climate change(s) through changes in species-level habitat suitability patterns. While changes in SR indicate climate change impacts on plankton alpha diversity, these do not inform us on the potential impacts on beta diversity (i.e., changes in community composition[81]). A community that experiences the replacement of all its constituting species by an equivalent number of newcomers will display a 100% rate of community turnover but no changes in SR. To investigate the amplitude of global plankton species turnover triggered by climate change, we examined future total turnover in annual species composition using Jaccard's dissimilarity index while decomposing its two additive components: nestedness (i.e., changes in SR) and true turnover (ST), which indicates the % of species that will be replaced in a community[27] using the betapart R package.

To do so, the mean annual species habitat suitability patterns used to estimate the ensemble changes in SR had to be converted to presence–absence maps as the Jaccard's dissimilarity index requires binary inputs[80]. A range of thresholds (0.10 to 0.80, by steps

of 0.01) was first explored for each SDM type (GLM, GAM, ANN, and RF) to infer threshold-based annual SR patterns. Then, we quantified the similarity of the threshold-based annual SR vs. the probability-based annual SR using Spearman's rank correlation coefficient ($\rho$) and ordinary linear regressions ($R^2$) to identify the range of thresholds that best match the probability-based estimates. The 0.25–0.40 range provided the most similar global SR patterns for GLMs, GAMs and ANNs (all $\rho > 0.95$, and all $R^2 > 0.90$). The 0.10–0.25 range was chosen for RF models. These ranges largely overlap with the species mean probability thresholds that maximize the TSS/AUC evaluation metrics, which are commonly used to convert habitat suitability into presence–absence maps. However, the maximizing-TSS approach tends to underestimate the natural gradual response of organisms to environmental variations, which is particularly problematic when dealing with SR patterns of widely-dispersed and climate-sensitive ectotherms such as the plankton[75–77]. Therefore, we chose to rely on a range of thresholds instead as it enables us to account for a wider range of possible realizations of community composition and better reflect the dynamic occupancy patterns inherent to planktonic taxa. Consequently, we derived ST estimates in annual species composition for each of the SDM-dependent threshold mentioned above and every of the 80 annual projections combinations, for total plankton, phyto- and zooplankton separately. Again, the ensemble projection in annual ST was derived by averaging those projections.

We further examined how climate change could impact not only community composition but also those species associations within the community that represent potential biotic interactions, which support ecosystem functioning[3,20,21]. Based on the mean annual species composition estimates used to compute ST rates, a text analysis algorithm[82,83] was used to identify pairs of species, which co-occur more frequently than expected given their individual occurrence. In short, the text analysis algorithm assigns an association score to each possible pair of two plankton species in all grid cells based on a likelihood ratio (LLR)[82]. The latter compares the probability of two species co-occurring together to the probability of one species occurring without their partner (i.e., two alternative probabilities), or when both are projected as absent, based on a combination of Shannon's entropy indices (H'). LLR values are >0 and they scale with the significance level of the projected species association, whether it is a positive (co-occurrence) or a negative (one-sided occurrence or co-absence) association. To disentangle between those two cases, when a species pair displayed an observed co-occurrence frequency lower than the product of the one-sided occurrence frequencies normalized to sample size, its LLR value was multiplied by −1. This way, we can identify those species pairs whose co-occurrence probability is lower than the products of the two one-sided occurrence probabilities (LLR < 0).

Again, for each probability threshold within the above-mentioned SDM-dependent ranges, and for each of the 80 annual projections, LLR values were inferred from the annual species composition estimates for both the contemporary and the future ocean. Negative LLR values and LLR values lower than the 75th percentile of the positive values were considered as non-significant species associations that are unlikely to represent potential species interactions. Therefore, we focus on those species pairs that present the highest positive LLR values in the contemporary and the future global ocean. Considering we cannot ascertain the direction or the nature of such biotic interactions from the literature or the data at hand, we only interpret them as strong species associations that represent potential species interaction at the scale of the global surface ocean. The number and the identity of those significant species associations were retrieved for the two time periods and compared for matching combinations of projections (e.g., contemporary annual composition based on p3 + ANN vs. future annual composition based on p3 + ANN + CESM1) to quantify how many remained constant and how many are lost or gained due to climate change. This way, we estimate how anthropogenic climate change might impair and/or reshuffle species interactions within the global plankton interactome.

## Estimating projections uncertainties and areas of non-analog conditions

The amount of variability around the final ensemble projections of %ΔSR and ST was measured through the standard deviation associated with the ensemble average. The amplitude and the spatial patterns of the standard deviation (Supplementary Fig. 4) indicate the level of uncertainties in our ensemble forecast projections[63,64]. Uncertainty is higher for phytoplankton SR projections (Supplementary Fig. 4) than for zooplankton SR (Supplementary Fig. 4), but, for both groups, the main spatial features of %ΔSR are conserved across all projections. Higher levels of uncertainty arise in regions where models disagree on the amplitude of the SR response to climate change. Indeed, for SDMs as well as ESMs, some models emerge as more sensitive than others, which leads to varying amplitudes in the %ΔSR predicted. Sensitivity ranking among SDMs (Supplementary Fig. 4) is as follows (from the least to the most sensitive): GLM < GAM < ANN < RF. Sensitivity ranking among ESMs (Supplementary Fig. 4) is as follows: GFDL-ESM2M < CNRM-CM5 and CESM1 < IPSL-CM5A-LR < MIROC5. The amount of uncertainty associated with the choice of the SDM algorithm is larger than the amount associated with the choice of the ESM, a feature commonly observed in analog studies[66,67].

Regions where climate changes lead to combinations of environmental predictors that have no analogs in contemporary conditions (i.e. novel climates) represent another source of uncertainty[84,85]. SDMs projections into novel climates might provide poorer forecasts if a model fits response curves that extrapolate in ecologically unrealistic ways (i.e., sharp exponential increase or decrease in habitat suitability). Here, we limit such

risks by providing the SDMs with background data that inform the potentially unfavorable habitats where a species could have been observed[69]. Furthermore, we took care to rely on SDMs of varying complexity and response shapes[70], which translate into varying sensitivity to climate changes (Supplementary Fig. 4). For instance, GLMs will provide smoother patterns when projected into novel conditions whereas RF will provide sharper transitions[70]. Relying on such differences provides alternative scenarios of how well species cope with non-analog conditions.

Nonetheless, we need to identify those cells where novel conditions emerge and estimate the level of extrapolation they could be associated with. Those cells were identified using the species Multivariate Environmental Similarity Surface (MESS[85]) algorithm from the *modEvA* R package. The MESS evaluates how dissimilar the environment is from the species' reference envelope (i.e., the SDM training set) used for SDMs training and generates a map presenting positive and negative values, the latter indicating the cells where extrapolation occurs. This method also identifies the environmental predictor(s) driving the extrapolation. As the outputs of the MESS depend on the calibration data (i.e., species occurrences and predictors), MESS values were computed at species-level for every set of predictors and for every possible monthly ESM projection, which represents 860 x 12 x 4 × 5 = 206,400 MESS estimates.

To summarize this information, mean annual MESS values were computed for each ESM separately and by distinguishing phyto- from zooplankton since their predictors set slightly differ. Mean annual MESS estimates indicate those regions of the future ocean where conditions that are outside the species' reference envelope occur on an annual scale. We also retained the frequency of each predictor being identified as a variable driving MESS < 0. Similar mean annual MESS patterns were found for the five ESMs (Supplementary Fig. 5): relatively low rates of MESS < 0 occur in the western tropical Pacific Ocean and in the Indian Ocean and those regions of non-analog conditions tend to expand with ESM sensitivity. The two most sensitive ESMs (IPSL-CM5A-LR and MIROC5, Supplementary Fig. 5) show more widespread and larger mean annual MESS patterns than other ESMs (Supplementary Fig. 5), and exhibit MESS < 0 in the tropical Atlantic and eastern tropical Pacific Ocean. MIROC5 projections (Supplementary Fig. 5) displays highly dissimilar conditions in some parts of the polar oceans where it predicts substantial increases in log(Chl). Otherwise, the majority of the future tropical dissimilarity were driven by SST increase outside the monthly SST ranges experienced by the species in the contemporary ocean.

## Spatial overlap between changes in plankton diversity and marine ecosystem services

We wanted to analyze regional patterns in the ensemble projections of plankton diversity changes and investigate how they might impair the contemporary provision of plankton-related marine ecosystem services[1]. To do so, we implemented a two-stage approach. First, the surface global ocean was clustered based on five variables that summarize our ensemble projections of climate change impacts on annual plankton diversity: percentage difference in phytoplankton SR (%ΔSR$_{phyto}$), percentage difference in zooplankton SR (%ΔSR$_{zoo}$), ST in phytoplankton specie composition (TO$_{phyto}$), ST in zooplankton species composition (TO$_{zoo}$) and total turnover between the future and the contemporary communities (mean Jaccard dissimilarity index). The latter corresponds to the ensemble projection based on the average Jaccard's dissimilarity index from which the ST estimates are derived (see section "Ensemble projections of changes in species richness, community composition turnover, and changes in species associations between the contemporary and the future ocean"). It represents the total turnover resulting from both SR and ST. Clustering the ocean based on these five variables enables us to identify regional hotspots of climate change impacts. We explored a variety of existing clustering approaches by first using nine different strategies based on alternative distance matrices, hierarchical or partitional (e.g., *k*-means like) approaches and varying linkage methods[86]. Five approaches were based on a Euclidean distance matrix and four were based on a Mahalanobis distance matrix[86]. Among Euclidean-based approaches the first consisted in performing k-medoids partitioning[87] based on a distance matrix computed from the untransformed ensemble projections. The four others consisted in performing a principal component analysis (PCA[86]) prior to the distance matrix computation to reduce the dimensionality of the data into uncorrelated synthetic components and smoothen the ensemble projections patterns. The Euclidean distance matrix was then computed from the scores of the first four principal components as those consistently explained more than 95% of total variance (PC1 = 62.2%; PC2 = 20.3%; PC3 = 12.6%, and PC4 = 4.1%). The second approach consisted in performing k-medoids on the latter. The third to fifth Euclidean-based approaches consisted in performing hierarchical clustering on the PCA-derived distance matrix based on three agglomerative linkages:[86] Ward's, average and complete respectively. The four Mahalanobis-based approaches followed the same plan but without the prior PCA as Mahalanobis distance computation involves the transformation of the variables into scaled uncorrelated ones. Again, k-medoids partitioning was performed on the first matrix and hierarchical clustering based on the three above-mentioned linkages constituted the three other approaches. For each of the nine approaches, 2 to 10 clusters were drawn and their robustness was examined through profiles of average silhouette widths and Calinski–Harabasz (C–H) indices (i.e., ratio of variances criterion)[88] to guide our choice of the optimal number of clusters. The silhouette and the C–H profiles converged towards four to six clusters for all approaches so we mapped the corresponding clusters and compared their variations in plankton diversity changes visually with boxplots. The final clusters choice was based on the C–H profiles and the clusters' spatial coherence, meaning that clusters providing regions that

were very imbalanced in terms of density and/or very scattered around the global ocean were discarded as they would be poorly informative. We narrowed our choice down to the Euclidean-based k-medoid partitioning among which we finally chose the six clusters from the PCA-based approach as it: (i) displayed the clearest C–H profile (i.e., C–H index was much higher for $k = 6$), and (ii) provided the clearest regional patterns. Finally, the scores of each grid cell along the four PCs were summed by weighting the scores by the percentage of variance explained of their corresponding PC (given above). The median absolute value (as PC scores can be negative) of this weighted sum constitute a continuous quantitative index summarizing the severity of climate change impacts on the plankton community within each of the six regions. The six regions were ranked in decreasing order this newly-defined severity index (Fig. S6). The main R packages used for the clustering approach were: FactoMineR, cluster, FD and fpc.

Second, various variables of major marine ecosystem properties were collected from the literature. Six variables representing proxies for plankton-related processes that provide crucial socio-economical services to human societies, from climate regulation through the sequestration of atmospheric $CO_2$, to tourism income and food provision[1,89] were retained: (i) normalized global species richness of oceanic taxa (bony fishes, sharks, cetaceans and squids)[13], which is indicative of overall marine biodiversity;[89] (ii) mean annual reported and unreported catch rates of small (<30 cm) pelagic fishes over the 1990–2019 period[34], which indicates the regions where large quantities of planktivorous fishes are collected; (iii) mean annual surface net primary production (NPP, mg Carbon m$^{-2}$ day$^{-1}$);[35] (iv) the corresponding flux of particulate organic carbon (FPOC, mg Carbon m$^{-2}$ day$^{-1}$) that is exported below the euphotic zone[35], which indicates the strength of the biological carbon pump (BCP); (v) the ratio of the two (e ratio = FPOC/NPP), which indicates the efficiency of the BCP; and (vi) the inverse of the mean annual slope of the power-law particles size distribution (PSD) measured from satellite ocean color observations[36], which is indicative of pelagic size structure and shows where larger organisms can emerge. Once these proxies were matched with our ensemble projections, we performed nonparametric variance analyses (Kruskal–Wallis tests[90]) to examine if they present significant variations across the regions defined above based on a 0.01 significance level. Normality and homoscedasticity tests were performed prior to the Kruskal–Wallis tests. When significant variations were found between regions, pair-wise post hoc tests of multiple comparisons of mean rank sums were performed to identify which pairs of regions displayed significant variations. Bonferroni corrections were applied for p-values adjustment. All variance analyses were carried out using the PMCMR R package. When comparing mean annual fisheries catch rates, region 5 (i.e., Southern Ocean and parts of the tropical upwellings) was discarded because the majority of low catch rates for the Southern Ocean are due to a lack of reported data. This way we test whether the contemporary provision of marine ecosystem services overlaps with regional hotspots of changes in plankton SR and community composition. By doing so we examine if anthropogenic climate change puts those services at risk through the reshuffling of plankton species composition. The distribution of the six regions, the spatial patterns of the variables used as well as the results from the variance analyses are reported in Supplementary Note 2.

## Examining diversity patterns and trends for major plankton functional groups (PFGs) and global structure of microphytoplankton cell size and zooplankton body size

While we cannot infer mechanistic links based on our ensemble modeling approach and from the data at hand, we overcome the gap between our estimates of species diversity and community composition and the variables of ecosystem services by: (i) examining global patterns and future trends in the species richness of ten PFGs[22], which represent groups of species that fulfill similar functions in food-webs and biogeochemical cycles; and (ii) examining global patterns and future trends in the size structure of diatoms and copepods, which are the two PFGs containing the most species modeled for phytoplankton and zooplankton, respectively. For both experiments, we make use of our ensemble projections of monthly habitat suitabilities to estimate annual patterns in SR for the various PFGs. Then, we computed Spearman's rank correlation coefficients to assess the strength of associations between the PFGs diversity, phyto- and zooplankton size structure, key environmental covariates (SST, logChla and nutrients concentrations), and the proxy variables of ecosystem service described above (Supplementary Note 9). First, all of the species modeled (Supplementary Data 2) were classified into a PFG based on the performance of particular biogeochemical functions (e.g., calcification vs. silicification[22]) and/or size classes that describe their trophic level in food-webs (e.g., nano-, micro-, or mesoplankton). Here, we opted for similar criteria and chose to focus on ten PFGs (three for phytoplankton and seven for zooplankton). Taxonomic classification served as strong basis for the present grouping, but some of the PFGs are paraphyletic and included taxa from either the same size class[91,92] and/or that are known to fill the same trophic niche as a result of similar feeding traits[92]. The PFGs investigated and their main functions/traits are:

*Diatoms* (n = 154): Microphytoplankton (20–200 μm) silicifiers (i.e., deplete silica concentrations) considered the main contributors to phytoplankton biomass, especially in cold and nutrients-enriched waters. This group is a key contributor to carbon export through diverse processes: high sinking rates through grazing, ballasting of large and mineralized cells, resting spores, etc.[4]).

*Dinoflagellates* (n = 154): Mixotrophic (i.e., use both organic and inorganic sources of carbon) nano- to microphytoplankton (2–200 μm) that produce and consume

phytoplankton biomass. Several Dinoflagellates are known to trigger harmful algal blooms[93]. Dinoflagellates are often considered as "gleaners" and are thus less competitive than the fast-growing Diatoms and some Coccolithophores under nutrients-replenished conditions[94].

*Haptophyta* (n = 23 Coccolithophores + Phaeocystis pouchetii): Nanophytoplankton (2–20 μm) calcifiers responsible for half of the marine $CaCO_3$ (calcite) fluxes that trigger large blooms at high latitudes, thus influencing marine primary production, alkalinity, carbonate and carbon chemistry on short to geological time scales.

*Copepods* (n = 272): Crustacean mesozooplankton (0.2–20 mm) considered to be the main grazers of phytoplankton, which they mainly capture through active current feeding, though some species do rely on alternative feeding-strategies[92,95]. Copepods strongly contribute to the biological carbon pump not only through phytoplankton grazing but also through the production of fecal pellets and by performing diel vertical migrations.

*Malacostraca* (n = 51 Euphausiids + 20 Hyperiid amphipods): Crustacean macrozooplankton (>10 mm) that graze on large phytoplankton and mesozooplankton through active filter feeding and cruising. Compared to mesozooplankton, Euphausiids (i.e., krill) display very high swimming speeds enabling them to perform larger vertical migrations, but they also show higher feeding rates leading to the production of very large fecal pellets. Euphausiids are a key component of biogeochemical cycles and trophic-webs in regions such as the Southern Ocean[96]. Hyperiids are also crustacean macrozooplankton (>2 mm) specialized in commensalism and parasitism of gelatinous zooplankton[97].

*Jellyfish* (n = 67 holoplanktonic Cnidaria + 2 Ctenophora): Gelatinous macrozooplankton (>20 mm) that perform passive ambush feeding to capture smaller zooplankton[92]. Large Jellyfish can form local outbursts of gelatinous biomass that efficiently export carbon[98,99].

*Chordates* (n = 11 Salpida and Doliolida + 6 Appendicularia): Barrel-shaped gelatinous meso- to macrozooplankton (0.5–200 mm) that radically differ from Jellyfish though they are often clumped as gelatinous zooplankton[100]. They are colony-forming passive filter feeders that pump water into their body where particles are retained on a mucous net. This strategy allows them to capture particles ranging from 1 μm to 1 mm whereas crustacean filter feeders (e.g., Copepods and Euphausiids) usually feed on particles <200 μm. Salps and Doliolids contribute to an efficient pathway of carbon export as they produce large and fast-sinking fecal pellets[100]. Appendicularians are part of the mesozooplankton (<10 mm) and perform passive filter feeding through a gelatinous house, which they use to aggregate particles[92]. Discarded Appendicularians' houses are a great source of food for detritivorous copepods and a potentially strong pathway of particles export[92,101].

*Chaetognaths* (n = 27): Exclusively marine group of translucent arrow-shaped worms that are found in the larger mesozooplankton (1–100 mm). They prey on smaller mesozooplankton (i.e., copepods) through active ambush feeding based on mechanoreception[92].

*Pteropods* (n = 19 Thecosomata): Calcifying mesozooplankton (<20 mm) that perform passive filter feeding through a mucus net that aggregates the particles they feed on (i.e., flux-feeding[92]).

*Foraminifera* (n = 26): Calcifying unicellular mesozooplankton (<1 mm[102]); characterized by calcareous shells with chambered perforated tests from which their ectoplasm can reach out to catch food items (i.e., passive ambush feeders). Many extant planktic Foraminifera also bear endophotosymbionts that make these organisms mixotrophic instead of purely heterotrophic[102]. Together with Coccolithophores and Pteropods they are the main producers of marine $CaCO_3$ in the marine plankton.

Second, we supplemented our analyses linking plankton species diversity and ecosystem functioning by combining the monthly species habitat suitability projections to a master functional trait, i.e., size: cell size for phytoplankton, and body size for zooplankton. Organism's size is a master trait because it transcends all major Darwinian functions (i.e. growth, feeding, survival, reproduction) and affects the expression of most other functional traits[103,104]. By using species composition patterns to estimate community-level size structures (i.e., the relative proportions of small and large species in an assemblage) we can better support and discuss our findings regarding the potential links between species diversity and ecosystem functioning. A full synthesis of cell size/body size accounting for all the 860 species modeled in our study remains to be implemented. However, extensive compilations of size measurements exist for two of the main PFGs investigated here: Diatoms[105] and Copepods[95,106]. These two PFGs alone represent 49,5% of the species studied here (Diatoms = 45.8% of phytoplankton species, Copepods = 51.9% of zooplankton species) and their diversity patterns are representative of the SR of their corresponding trophic level (Supplementary Note 6, Fig. 1).

For Diatoms, we retrieved the species-level measurements of average cell volume (μm³), average cell surface to volume ratio (S/V, μm$^{-1}$) and average cell Carbon (C) content (mg C m$^{-3}$) documented in ref. [105]. Diatom species names were carefully examined and corrected to find the matching species labels defined in ref. [40]. Then, species-level (n = 534) and genus-level (n = 135) mean estimates of cell volume, S/V and C content were computed from every measurement available in ref. [105]. Species displaying <5 cell measurements were given genus-level mean estimates (n = 13, 8.4% of Diatom species modeled). Two Diatom species displaying a mean S/V > 10 were considered as artifactual outliers and discarded from the dataset. Ultimately, all 154 Diatom species modeled could be attributed a mean estimate of cell volume, S/V and C content. Then, for the contemporary ocean, the monthly projections of Diatom species habitat suitability indices (HSI) from each model member (n = 16) were combined with the species-level estimates of average cell size to estimate monthly HSI-weighted median Diatom cell

volume, $S/V$ and C content. The diversity of Diatom cell volume estimates, $S/V$ and C content was estimated through the corresponding HSI-weighted variance. Each monthly model member estimate was then used to derive an annual median estimate for all these six variables aimed to characterize the various dimensions of the Diatom community size structure emerging from our HSI projections (Supplementary Note 10). Our estimates of global Diatom community cell size structure were compared to the satellite-based estimates of global phytoplankton size from ref. [36] for validation (Supplementary Note 10). Copepod body size (mm) data were used in a similar fashion. The body length measurements of adult females reported in ref. [43] and summarized in refs. [95,106] were retrieved. Mean maximal body length was computed for every planktonic copepod species with >5 measurements. A mean maximum body size could be attributed to 91.5% of copepod species ($n = 249$, 47.5% of all zooplankton species modeled). Again, the monthly projections of Copepod species HSI from each model member ($n = 16$) were combined with the estimates of average body size to derive monthly HSI-weighted median Copepod body size. Copepod body size diversity was estimated through the corresponding HSI-weighted variance. Every monthly model member estimate was then used to derive an annual median estimate for these two variables that characterize the size structure of surface Copepod communities, in the surface contemporary open ocean (Supplementary Note 11). Our estimates of global Copepod community size structure were compared to the previous global model estimates of ref. [107] for validation.

Finally, we assessed the impact of future climate change on Diatom and Copepod size structure in the same way as for species diversity. The future monthly HSI projections of Diatom and Copepod species from each model member ($n = 80$) were combined with the species-level estimates of cell size and body size described above to estimate the microphytoplankton and zooplankton community structure patterns emerging in the future ocean. By looking at the % difference between the future and the baseline estimates of community size structure, we investigated how climate change might reshuffle the size structure of the phyto- and zooplankton communities. More specifically, we tested whether future warming will lead to a decrease in median cell volume (and thus an increase in cell $S/V$) and body size through the replacement of larger species by smaller ones, a process expected under global warming, particularly towards higher latitudes[2,6,37,108]. All of the results associated with section "Examining diversity patterns and trends for major plankton functional groups (PFGs) and global structure of microphytoplankton cell size and zooplankton body size" are thoroughly summarized in Supplementary Notes 9 to 11.

The authors checklist, the main hypotheses and models assumptions and every key step of our species distribution modeling approach are summarized in a standard Overview, Data, Model, Assessment and Prediction (ODMAP) protocol[109] to ensure the traceability and reproducibility of our study (Supplementary Note 5).

**Reporting summary**. Further information on research design is available in the Nature Research Reporting Summary linked to this article.

## Data availability

The gridded spatial fields of contemporary and future plankton/phytoplankton/zooplankton species diversity estimates generated in this study were deposited in Zenodo [10.5281/zenodo.5101518]. The phytoplankton species occurrences data used to model global phytoplankton species diversity and composition are available on PANGAEA [10.1594/PANGAEA.904397]. The zooplankton species occurrences data used to model global zooplankton species diversity and composition are available on Zenodo [10.5281/zenodo.5101349].

## Code availability

Any computer code used to generate the results of the study are freely available upon request to the authors and all R codes are currently stored on the GitHub account of F.B. (https://github.com/benfabio).

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

## Acknowledgements

We thank all contributors involved in the plankton species field sampling and identification and we acknowledge the efforts made to share such data through publicly available online archives such as OBIS, GBIF, and MAREDAT. We thank the CMIP5 and MAR-EMIP initiatives as well as Thomas Frölicher and Charlotte Laufkötter for facilitating the processing of the earth system models outputs. We thank Federico Ibarbalz for kindly sharing his model projections with us. We also thank the Environmental Physics group for providing feedback on earlier versions of the study and particularly to Luke Gregor for improving the writing. F.B. received support from ETH Zürich. This project has received funding from the European Union's Horizon 2020 research and innovation programme under grant agreement No 862923. This output reflects only the author's view, and the European Union cannot be held responsible for any use that may be made of the information contained therein. D.R. was supported by ETH Zürich under grant ETH-52 13-2.

## Author contributions

F.B., M.V., and N.G. conceived the study. F.B. collated and analyzed the data with support of M.V., N.G., D.R., and U.H.E. F.B., M.V., and N.G. wrote the manuscript, with inputs from N.E.Z, D.R., and U.H.E.

## Competing interests

The authors declare no competing interests.
