## [Peer Review File · Nature Communications]

Reviewers' Comments:

Reviewer #1:

Remarks to the Author:

In their manuscript, Benedetti et al explore the potential consequences of GC on oceanic phyto- and zooplankton diversity. They evaluate future plankton diversity under the so-called RCP8.5 scenario. Their analysis is based on species distribution models. In general, they find poleward movements of peak diversity, with some pronounced differences between phyto- and zooplankton. The paper contributes some interesting forecasts reg changes in plankton diversity. However, the results seem a bit inconclusive without reporting consequences on functional aspects of plankton diversity and ecosystem functioning. The text os often hard to read, using too many abbreviations which are not introduced. Overall it seems to fit better to a more specific journal with a focus on modeling. I was also missing some transparent validation of the SDMs, e.g. by comparing the SDMs along temperature/latitudinal gradients. As very non-intuitive result, the authors find that changes in phyto diversity respond more to changes in temperature than resources or water column stability. Moreover, they report overall increasing phyto diversity with GC exc $>70^{\circ}$ North. Both of these predictions seem very no-intuitive as (a) resource availability is crucial for producer diversity and (b) SST is expected to increase especially polewards.

Specific comments:

Abbreviations not introduced before usage, e.g. MIROC5, CIMP5, RCP8.5

l.74 "habitat suitability model_s_"

l. 74-76 "using results from five ESMs part of the CMIP5 project [25] that were run following the high emissions scenario RCP8.5 [25]." –pls formulate complete sentences

l. 88 – a reference how you separate nestedness from replacement?

l.126 "this is the first global study to project the future SR for phytoplankton, revealing that the two trophic levels will experience rather different trajectories" – phytoplankton is one trophic level

Reviewer #2:

Remarks to the Author:

The paper of Benedetti and collaborators provides substantiated evidence of the changes in plankton species richness and community composition determined by global warming.

The study also anticipates future changes and possible knock-on effects on plankton-mediated ecosystem services, under a high emission "business as usual" scenario.

The authors here develop a set of Species Distribution Models, which are validated using the best information available following rigorous methodological procedures and using appropriate numerical tools. These models are then run to forecast changes in the plankton compartment at the end of this century, based on projections of future environments obtained by different Earth System models.

Overall the paper of Benedetti et al provides a significant contribution towards an improved understanding of the impact of global warming on the marine environment, raising awareness on the implications associated with highly relevant ecosystem services. Ensemble Species distribution models here developed are improved in comparison with previous models, as they take into account a relative high number of environmental predictors and are validated using a very extensive data set following robust methodological procedures.

A major claim of the paper is the identification of different trajectories of changes for phyto- and zooplankton, which raises new awareness around the complexity of changes that might take place at the end of this century if nothing is done to reduce current carbon emissions.

In conclusion, this is an important paper that deserves to be published in Nature Communication, after that a few minor concerns are carefully addresses.

In particular it should be further clarified in the abstract and discussion that results here presented mainly focus on offshore plankton species (i.e. species distributed beyond 200m depth) and that

consequently the anticipated effects on plankton-mediated ecosystem services, which are often located in demersal waters, could have been underestimated. Also the comprehensive results here presented are sometimes difficult to follow and in particular some figures would benefit from a few minor amendments.

Below is reported a list of minor concerns that should be addressed before publication, while others are directly annotated in the text and supplementary material.

Minor changes- text

- Please clarify how it is possible that the average maximum sampling depth is 73 m depth (line 554), when the occurrences of species associated with seas shallower than 200m were removed (lines 531-532)
- Please possibly clarify why meroplanktonic species of jellyfish were eliminated (lines 546-548), while other meroplanktonic taxa such as polychaetes were retained;

Minor changes- Figures

Figures' legends are overall quite heavy to follow. Therefore it would be helpful if also supplementary figures could have the main title highlighted in bold (as done in Figures 1-4).

- Fig. 4 and its legend need to be carefully revised to further clarify the results therein presented. In particular in fig. 4b the units should be indicated in the title of each graphs, replacing the numbers in brackets that refers to bibliographic references, which are already indicated in the text. The legend of figure 4 should include an explanation of the figure shown in the right panel, to be indicated as Fig. 4c. A possible text could be something along the following lines:
Figure 4: Distribution of (a) the ocean regions defined and ranked according to the severity of climate change impacts on their plankton community and (b) how they overlap with the contemporary provision of marine ecosystem services. The median and upper/lower quartiles of the ranking indices in the different regions and a summary of the main plankton changes and hypothesized impacts on the associated marine ecosystem services are also shown (c).
- In Figs. S2-3 and S2-6 the titles of the x-axes, as well as chart titles above and to the right of the graphs are unreadable. Bigger titles and a larger legend showing environmental predictors in different colours should suffice to describe the box-plots here presented. In whatever way these figures are revised, the names of different phyto-/zooplankton groups and environmental predictors should be made readable.

See also additional comments and notes at:

- pag. 7, 18, 19, 27, 28, in the file with the main text;
- pag. 4, 7, 8, 13, 21, 22, 24 in the file with supplementary material;

COMMENTS TO THE AUTHORS OF MANUSCRIPT NCOMMS-20-37764-T WITH RESPONSES FROM THE AUTHORS (IN BLUE)

1. COMMENTS FROM REVIEWER#1

- A. In their manuscript, Benedetti et al. explore the potential consequences of GC on oceanic phyto- and zooplankton diversity. They evaluate future plankton diversity under the so-called RCP8.5 scenario. Their analysis is based on species distribution models. In general, they find poleward movements of peak diversity, with some pronounced differences between phyto- and zooplankton. The paper contributes some interesting forecasts reg changes in plankton diversity.

Authors: We would like to thank reviewer#1 for taking the time to review our study. We hope that the revisions made in the present version will address his/her concerns regarding the more functional aspects in our results and conclusions (see below).

However, the results seem a bit inconclusive without reporting consequences on functional aspects of plankton diversity and ecosystem functioning.

Authors: We agree that our previous version did not address functional aspects very clearly. At the same time, reviewer #2 found that our study provides “a significant contribution towards an improved understanding of the impact of global warming on the marine environment, raising awareness on the implications associated with highly relevant ecosystem services”. This may reflect highly different perspectives on what is important when analyzing/assessing global change impacts. Nonetheless, we agree with reviewer #1 that adding aspects related to ecosystem functioning strengthens the manuscript. For the revised version of our study, we carried out several in-depth analyses related to functional responses. .

Specifically, we strengthened our study by looking at diversity patterns and future trends for ten different Plankton Functional Groups (PFGs, see response below plus new section E.4 of the Methods) that are known to fulfill distinct biogeochemical and/or trophic functions in marine ecosystems. We now document their latitudinal diversity gradients (which further validate our global diversity patterns, see Document S1 and response below) and how they respond to warming individually (Document S1 and S16). In addition, we document how the richness of these PFGs relate to ecosystem services (Document S16). In addition, we added a size-based approach for the two most-studied PFGs of each trophic level (diatoms for microphytoplankton and copepods for zooplankton) which enabled us to better link plankton species diversity to the variables of ecosystem functioning. We made use of published species-level cell volume (but also cell surface to volume ratio and cell carbon content) and body size measurements to derive monthly and annual median estimates of diatom and copepod community size structures from our ensemble model members (Document S17 and S18) that reflect where a higher proportion of large taxa occurs relative to smaller ones. Our estimates are compared to previous estimates and we discuss their limitations and how to interpret them in the Supplementary Documents. Considering that species size is viewed as a ‘master’ functional trait that transcends many ecosystem functions, and that it is one of the few traits available in the literature for such a broad range of species (and retrievable within the scope of revisions for >150 diatoms and copepod species), we believe our new analyses link our estimates of species

composition better to the proxy variables of ecosystem services. This way, we used our species diversity projections to infer sensible latitudinal patterns of plankton size (which would further strengthen the validity of our model members) and tested whether our projected warming-driven shifts in species ranges conform to the view that anthropogenic climate change favors small organisms at the expense of larger ones, with notable implications for ecosystem production and carbon export (but see literature cited throughout the manuscript). We believe our new analyses in response to the request by reviewer #1 greatly enhanced the quality of our study and support our initial results. We thank the reviewer for giving us the opportunity to strengthen this major aspect of our study, and we hope s/he will find that our revisions addressed his/her main concerns.

The text is often hard to read, using too many abbreviations which are not introduced.

Authors: We are sorry the reviewer found some parts of the text hard to read. We tried to clarify the text in the revised version. There were indeed several acronyms that were not introduced properly. We made sure to define these in the revised version (e.g. lines 77-79 + section D of the methods).

Overall it seems to fit better to a more specific journal with a focus on modeling. I was also missing some transparent validation of the SDMs, e.g. by comparing the SDMs along temperature/latitudinal gradients.

Authors: We feel we did a sound job in testing SDMs, as we employed classical procedures used in the SDM literature and by comparing the derived patterns along latitudinal gradients (which was also acknowledged by reviewer #2). In addition, we assessed in our manuscript (Fig. 1a,c,e and Fig. S13) patterns of agreements between SDM methods along latitudinal gradients. Model evaluation tests were thoroughly evaluated and validated using state-of-the-art protocols that were adapted to the nature of sparse plankton occurrence data (Fig. S11). In section C of the Methods (lines 632 to 887), we ensured the transparency and rigor of our methods by providing extensive details regarding every step involved in SDMs implementation (sections C1, C2 and C3), projection (section C4), and how we cared to address all the main sources of uncertainties inherent to such approach (sections C1, C2, C3.1, C3.2, C.5 and then E.2). Additionally, we provided several supplementary information that support our methodological choices and document how we dealt with major uncertainty sources: Doc. S2 extensively documents our comprehensive framework of predictors selection; Doc. S4 shows how latitudinal variations in sampling effort (i.e. gap in occurrences near the equator) do not drive the global zooplankton diversity pattern; Doc. S9 shows how different background data selection method weakly affect the modelled global zooplankton diversity pattern; Doc. S10 documents how we further tested for the impact of environmental predictors that were inducing biases in niche space for phyto- and zooplankton; Fig. S13 illustrates the quantitative level of uncertainties in our future projections and how they are driven by variability between species distribution models and earth system models; and Fig. S14 highlights those regions of the global ocean where the species distribution models are likely to be projected into environmental conditions outside of their calibration range (i.e. non analog conditions). Those regions were highlighted with hatching on Figure 1 and identified following a very standard method in the SDM community.

Regarding the global phytoplankton diversity patterns, a direct comparison with independent data is not possible on a global scale (as explained in Righetti et al. 2019) due to lack of sufficient available data. Yet, we would like to stress that the present diversity patterns are highly consistent with those of Righetti et al. (2019), in spite of some differing decisions with regards to the number of species modelled (l. 860-866 in the Methods) and the process of environmental predictors selection

(although both led to very similar predictors being selected in the models, ultimately), as these steps had to be adapted due to the limited predictors available for future climate change projections. Our phytoplankton baseline global richness patterns were directly compared to those obtained by Righetti et al. (2019) for various background selection procedures (total target group background versus group level-target group background) through non parametric correlation tests (Spearman rank correlation, rho), after normalizing to the total number of species modelled. All tests yielded significant correlations coefficients ranging between 0.89 and 0.91 (all p-values < 0.001). This confirms the robustness of our approach to the methodological choices mentioned above.

In summary, we believe that our procedures to evaluate and assess models statistically and along global latitudinal gradients are robust. Nonetheless, we further strengthened our case. In the revised version, we now show global maps and latitudinal patterns (sensu Figure 1) for the various PFGs considered here (see Fig. S1) and compare them where possible with previous independent studies that documented such patterns based on observations. This way, we further support our findings by explicitly showing how our approach successfully models plankton group diversity patterns. We ask reviewer #1 to bear in mind that most previous studies are based on regional, or partially global, cruises so their ensuing diversity maps likely capture smaller-scale patterns than those covered by our global approach (local points vs. global 1°x1°, seasonal vs. mean annual across >30 years of observations, abundance vs. occurrence only). We list below the previous observations-based studies that were used as references for the Latitudinal Diversity gradients (LDG) of the various PFGs (some of them were also used to validate the diversity-SST relationships we found in Fig. 2):

- **Diatoms:** Olguin Salinas et al. (2015), Ibarbalz et al. (2019), Busseni et al. (2020).
- **Dinoflagellates:** no previous study documenting global marine Dinoflagellates LDGs; see Ibarbalz et al. (2019) and the LDG observed for marine phototrophic/mixotrophic Protists.
- **Haptophytes** (mainly Coccolithophores): O'Brien et al. (2016).
- **Copepods:** Rombouts et al. (2010), Beaugrand et al. (2012), Hirai et al. (2020).
- **Malacostraca** (mainly Euphausiids): Tittensor et al. (2010).
- **Jellyfish:** no previous study documenting global marine Chordates LDGs
- **Chordates:** no previous study documenting global marine Chordates LDGs
- **Chaetognatha:** Miyamoto et al. (2014).
- **Pteropods:** Burrige et al. (2016).
- **Foraminifera:** Rutherford et al. (1999), Tittensor et al. (2010), Yasuhara et al. (2012).

We hope that this extension increases the confidence (of reviewer #1) in our results. All the necessary information was thoroughly reported in the revised Supplementary Document S1 (Fig. S1-2).

As very non-intuitive result, the authors find that changes in phyto diversity respond more to changes in temperature than resources or water column stability. Moreover, they report overall increasing phyto diversity with GC exc >70°North. Both of these predictions seem very no-intuitive as (a) resource availability is crucial for producer diversity and (b) SST is expected to increase especially polewards.

Authors: There are several reasons why our results are not necessarily “non-intuitive” (at least from our personal viewpoint).

First, we would like to highlight that we estimate phytoplankton diversity as a property emerging from the stacking of species spatial distributions that are constrained by temperature as well as several other variables that reflect various dimensions of macronutrients (NO_3 , SiO_2 , N^* , Si^*) and light availability (PAR). Because *in situ* observations that are comparable in terms of sampling methods are very scarce and would only allow to model a very limited number of taxa, all the plankton observations used in the present study were converted to presence only data. Therefore, the relative importance ranks of environmental predictors we estimated here (thoroughly reported in Supplementary Document S2) quantify the importance of drivers of distribution/range sizes, not species abundance or biomass which are indeed primarily controlled by resource availability which limits phytoplankton growth. Overall, SST emerges as the most important variable constraining global distributions across all species included in this study. This is in line with the previous study by Righetti et al. (2019) and in line with the general view of the metabolic theory of ecology. Therefore, it is consequential that we find it to be the strongest covariate of the emerging phytoplankton diversity pattern both under current conditions and especially also under changing temperature regimes. However, we would like to point out that PAR (i.e. light availability), N^* and Si^* (i.e. excess of nitrates and silicic acid, respectively) also rank as prominent constrains for the distribution of individual phytoplankton species (Supplementary Document S2). Hence, we are not postulating that resource availability is not important for phytoplankton diversity, but that SST has a stronger impact given both the analyzed distribution data and the projected predictors (see second point below). As said: these results are based on species ranges rather than species abundances or biomass or size (which we think is what the reviewer may be referring to). For the latter, it is quite likely that resources availability plays a more important role than SST. Our findings are further in line with several observation-based studies that often found sea surface temperature to be the main driver (or the co-main driver) of protists diversity in the global ocean (Rutherford et al., 1999; Brun et al., 2015; O'Brien et al., 2016; Ibarbalz et al., 2019; Bussen et al., 2020).

Second, we would also like to point out that phytoplankton species richness responds more strongly to future changes in temperature because temperature-related predictors (SST and dSST) are those projected to vary the most in relative amplitude as a result of anthropogenic climate change, based on our ensemble of earth system models. In the figure panels below, we illustrate the distribution of “% difference” (future - baseline) across the predictors used for our future projections, on a global scale (A) and per latitudinal bands (B). The graphs show that SST and dSST show stronger future changes than the other predictors. This fact, combined with the relatively higher importance of SST in constraining the SDMs, explains why temperature changes were found to be the main drivers of future changes in plankton diversity (especially in polar and temperate latitudes).

Third, we would also like to highlight that, contrary to what the reviewer seems to assume, we do not find a positive linear relationship between phytoplankton diversity and SST. As clearly shown in Fig. 2, phytoplankton richness can decrease with increasing temperature in cold areas. As shown in Fig. S15 of the present appendices, and thoroughly commented in Righetti et al. (2019), the relationship between mean annual temperature and phytoplankton richness is a non-linear one as seasonality in temperature can decrease the emerging mean annual species richness by selecting for fewer tolerant species. This is what is driving the slight decrease in phytoplankton richness that we project in the Arctic Ocean north of 70°N. Additionally, the areas >70°N also happen to constitute a region where our model members seem to disagree on the sign of the future changes in phytoplankton richness, as illustrated in Fig. 1i,j (< 50% of all 80 model members do not agree on the sign of changes in phytoplankton richness).

Overall, this is why we strongly believe our results are not “non-intuitive” and that we provided all the necessary material needed for readers to understand the factors underlying our future projections. Nonetheless, we agree with the reviewer that: (i) we should make it clearer in the text that we do not believe temperature to be the one and only driver of phytoplankton diversity, and (ii) we should emphasize that other drivers (related to resources availability) were found to be important drivers of species spatial distributions as well. We have addressed the reviewer’s comment by adding the following sentence in the main text (lines 180-184): “At the scale of our study, we found variables related to water mixing (e.g. wind stress and mixed layer depth) and resource availability (e.g., nutrients concentrations, surface irradiance, oxygen concentration) to be slightly weaker predictors of species distributions compared to SST although they still often rank among the top predictors for many functional groups (Supplementary Document S2).”.

B. Specific comments:

Abbreviations not introduced before usage, e.g. MIROC5, CMIP5, RCP8.5

Authors: We are sorry we forgot to introduce those acronyms properly in the text. We corrected the main text to define the CMIP5 and RCP abbreviations. We thoroughly clarified the name of each Earth System Model in the appropriate section of the Methods (section D, from line 937).

l.74 “habitat suitability model_s_”

Authors: We corrected the plural form accordingly.

l. 74-76 “using results from five ESMs part of the CMIP5 project [25] that were run following the high emissions scenario RCP8.5 [25].” –pls formulate complete sentences

Authors: We modified this sentence to introduce the abbreviations when needed. The new sentence now reads: “Assuming niche conservatism, we projected each of the 16 resulting habitat suitability models into the future using outputs from five ESMs belonging to the Coupled Model Intercomparison Project 5 (CMIP5, [25]) that were forced by the Representative Concentration Pathway 8.5 (RCP8.5, [25]) scenario of high greenhouse gas concentrations.”.

l. 88 – a reference how you separate nestedness from replacement?

Authors: Yes, this corresponds to Baselga (2010) which was previously referred as reference #81 in section E of the Methods. We have moved this reference to line 93 of the main text.

l.126 “this is the first global study to project the future SR for phytoplankton, revealing that the two trophic levels will experience rather different trajectories” – phytoplankton is one trophic level.

Authors: We are sorry that we employed a rather confusing phrasing. By “two trophic levels”, we actually meant both phyto- and zooplankton, which are commonly accepted as the first two trophic levels of marine food-webs. We clarified the sentence as follows (now lines 136-138): “this is the first global study to project the future SR patterns for phyto- and zooplankton in a comparable fashion, and it demonstrates that the SR of the two trophic levels might experience different trajectories”.

2. COMMENTS FROM REVIEWER#2

- A. The paper of Benedetti and collaborators provides substantiated evidence of the changes in plankton species richness and community composition determined by global warming. The study also anticipates future changes and possible knock-on effects on plankton-mediated ecosystem services, under a high emission “business as usual” scenario. The authors here develop a set of Species Distribution Models, which are validated using the best information available following rigorous methodological procedures and using appropriate numerical tools. These models are then run to forecast changes in the plankton compartment at the end of this century, based on projections of future environments obtained by different Earth System models. Overall the paper of Benedetti et al provides a significant contribution towards an improved understanding of the impact of global warming on the marine environment, raising awareness on the implications associated with highly relevant ecosystem services. Ensemble Species distribution models here developed are improved in comparison with previous models, as they take into account a relative high number of environmental predictors and are validated using a very extensive data set following robust methodological procedures. A major claim of the paper is the identification of different trajectories of changes for phyto- and zooplankton, which raises new awareness around the complexity of changes that might take place at the end of this century if nothing is done to reduce current carbon emissions. In conclusion, this is an important paper that deserves to be published in Nature Communication, after that a few minor concerns are carefully addressed.

Authors: We thank reviewer #2 for her/his very positive comments about our study. We hope that the revised version of the manuscript will satisfy all of the issues raised.

In particular it should be further clarified in the abstract and discussion that results here presented mainly focus on offshore plankton species (i.e. species distributed beyond 200m depth) and that consequently the anticipated effects on plankton-mediated ecosystem services, which are often located in demersal waters, could have been underestimated.

Authors: We agree with the reviewer that this key point should be made clearer for the readers. We now highlight this point in the Abstract and Discussion sections. As explained in the Methods section, we chose to focus on modelling holoplanktonic species that mainly occur offshore because: i) they are the ones contributing the most to open ocean plankton communities; and ii) their spatial ranges can be effectively modelled by the predictors selected, contrary to species that mainly occur along the coast and whose distribution is controlled by the very complex and

small-scale shelf processes. Indeed, as a result, our study does not cover the demersal waters.

Also the comprehensive results here presented are sometimes difficult to follow and in particular some figures would benefit from a few minor amendments.

Authors: We amended the figures according to the suggestions made by reviewer #2 (see below). We hope our revisions will make the figures better readable and comprehensible.

- B. Below is reported a list of minor concerns that should be addressed before publication, while others are directly annotated in the text and supplementary material.

Minor changes- text

-Please clarify how it is possible that the average maximum sampling depth is 73 m depth (line 554), when the occurrences of species associated with seas shallower than 200m were removed (lines 531-532)

Authors: We would like to clarify that the sampling depth refers to the depth that the samples were taken and not to the water depth at the location of these samples. Since the vertical sampling is often determined by instruments and logistics (e.g. standard sampling protocols usually tow plankton nets within the first 200m), the mean sampling depth across all samples is a priori independent of the seafloor depth. It is influenced, though, by the fact that we are analyzing observations only from stations that have a water depth of more than 200 m. Recall, also that we have discarded any occurrence associated to a sampling depth deeper than 500m to avoid accounting for deep dwelling species while still accounting for species that can perform deep diel vertical migrations. .

The revised part of the Methods section now reads: "To restrict observations to those occurrences collected in the environmental conditions prevailing in the euphotic zone, or the mixed layer, we discarded occurrences sampled with a net tow whose maximal sampling depth was >500m. The average depth was used when maximal depth was not provided in the metadata. Therefore, the maximal depth of a zooplankton species occurrence allowed is 500m. This way, we tried to account for the zooplankton community that frequently performs diel vertical migration across the euphotic zone or the mixed layer, and that often co-occurs with species inhabiting surface layers".

-Please possibly clarify why meroplanktonic species of jellyfish were eliminated (lines 546-548), while other meroplanktonic taxa such as polychaetes were retained

Authors: In line with our second response above, we chose to discard zooplankton species that present a meroplanktonic life cycle (i.e. alternation between a larval planktonic stage and a fixed adult benthic stage) to avoid accounting for occurrences of species that do not contribute to the pelagic plankton community. Thanks to the review by Gibbons et al. 2010 (reference #46) we were able to identify the Hydrozoan species (i.e. jellyfish) with such a life cycle. It is true that many marine Annelida present an adult fixed stage, but this is not the case for the four Phyllococida families (Tomopteridae, Alciopidae, Lopadorrhynchidae, Typhloscolecidae; represented by 11 species) that we considered here as they are all holopelagic (see the review by Halanych et al., 2007; <https://doi.org/10.1093/icb/icm086>). In this case, and contrary to Hydrozoans, we carefully downloaded occurrences at the family level to avoid having to discard the numerous meroplanktonic annelid taxa. The Methods section was clarified according to the reviewer's comment, and now reads as: "Prior to retrieving the

occurrence data online, we first identified the phyla (Order/Class/Family) that comprise the bulk of extant oceanic zooplankton communities: Copepoda (i.e. appendicularians), Ctenophora, Cubozoa (i.e. “box jellyfish”), Euphausiidae (i.e. krill), Foraminifera, Gymnosomata (i.e. “sea angels”, pteropods), Hydrozoa (i.e. jellyfish), Hyperiidia (i.e. amphipods), Myodocopina (i.e. ostracods), Mysidae (i.e. small pelagic shrimps resembling krill), Neocopepoda, Podonidae and *Penilia avirostris* (i.e. cladocerans), Sagittoidea (i.e. chaetognaths), Scyphozoa (i.e. jellyfish), Thaliacea (i.e. salps, doliolids and pyrosomes), Thecosomata (i.e. pteropods), and four families of pelagic Polychaeta (i.e. worms) that are often found in the zooplankton and whose species are known to display holoplanktonic lifecycles (Tomopteridae, Alciopidae, Lopadorrhynchidae, Typhloscolidae).”

Minor changes- Figures

Figures’ legends are overall quite heavy to follow. Therefore it would be helpful if also supplementary figures could have the main title highlighted in bold (as done in Figures 1-4).

Authors: The main captions of the Supplementary Figures were simplified and highlighted in bold in accordance with the reviewer’s comment.

- Fig. 4 and its legend need to be carefully revised to further clarify the results therein presented. In particular in fig. 4b the units should be indicated in the title of each graphs, replacing the numbers in brackets that refers to bibliographic references, which are already indicated in the text.

Authors: The bibliographic reference numbers were replaced by the units of the ecosystem services proxy variables.

The legend of figure 4 should include an explanation of the figure shown in the right panel, to be indicated as Fig. 4c. A possible text could be something along the following lines:

Figure 4: Distribution of (a) the ocean regions defined and ranked according to the severity of climate change impacts on their plankton community and (b) how they overlap with the contemporary provision of marine ecosystem services. The median and upper/lower quartiles of the ranking indices in the different regions and a summary of the main plankton changes and hypothesized impacts on the associated marine ecosystem services are also shown (c).

Authors: We thank Reviewer #2 for this suggestion. We modified the labels and the caption of Fig. 4 in line with his/her comment. The new caption reads as follows: “Figure 4: Distribution of (a) the ocean regions defined and (b) ranked according to the median severity index of climate change impacts on their plankton community and (c) how they overlap with the contemporary provision of marine ecosystem services. The regions were defined by clustering every raster cell of the global ocean based on their average projected difference in annual phyto- and zooplankton species richness, phyto- and zooplankton species true turnover and total plankton turnover. Six proxy variables linked to marine ecosystem services across the six regions were considered: oceanic megafauna biodiversity (SR) [13] (normalized species richness), mean annual catch rates of small (< 30cm) pelagic fishes (log(tons km⁻² yr⁻¹)) [33], annual net primary production (NPP; mgC m⁻² d⁻¹) [34], the corresponding fraction of particulate organic carbon exported below the euphotic zone (FPOC; mgC m⁻² d⁻¹) and the corresponding efficiency of the production exported (FPOC/NPP), and an index of mean annual plankton size [35]. (b) also summarizes how the changes in

plankton richness and composition might impact the marine ecosystem services shown in (c). More details are provided in Supplementary Figure S6.”

- In Figs. S2-3 and S2-6 the titles of the x-axes, as well as chart titles above and to the right of the graphs are unreadable. Bigger titles and a larger legend showing environmental predictors in different colours should suffice to describe the box-plots here presented. In whatever way these figures are revised, the names of different phyto-/zooplankton groups and environmental predictors should be made readable.

Authors: We agree with the reviewer that the labels of these very large panels of boxplots were hard to read. We changed the format of these figures and enlarged the labels and legends for better readability.

See also additional comments and notes at:

- pag. 7, 18, 19, 27, 28, in the file with the main text;
- pag. 4, 7, 8, 13, 21, 22, 24 in the file with supplementary material.

Authors: We thank the reviewer for going through our manuscript so carefully. All of these relatively minor comments were carefully addressed in the revised version of our manuscript. They are visible through the manuscript tracking.

Reviewers' Comments:

Reviewer #2:

Remarks to the Author:

I have read through the revised version of the MS of Benedetti et al, and I am fully satisfied by the way in which the authors have addressed my comments and minor concerns.

Overall I find the present version of the MS further improved in terms of content and clarity. I therefore confirm my previous evaluation, i.e. this paper provides a significant contribution towards an improved understanding of the impact of climate change on the marine environment and it certainly deserves to be published in Nature Communications.

Reviewer #3:

Remarks to the Author:

REVIEWER'S REPORT FOR

"Major restructuring of marine plankton assemblages under global warming." for Nature Communications.

by Benedetti Fabio and co-workers

In this MS, the authors use SDMs to determine changes in open ocean phytoplankton species richness and community composition under a high emission scenario (RCP8.5). Their analysis is based on a total of 860 species (336 phytoplankton and 524 zooplankton) from 10 plankton functional groups. They project future habitat suitability patterns for the present (2012-2031) and the end of the century (2081-2100) according to outputs of five Earth System Models, and examine changes in alpha diversity (species richness) and beta diversity (species turnover). Overall, their results indicate an increase in phytoplankton SR by 16% over most regions for the end of century, whereas zooplankton richness is predicted to increase strongly in temperate to subpolar latitudes, but to decline in the tropics, with major changes in species turnover pointing to a major community restructuring driven by poleward range shifts.

The study touches upon a critically important subject, given the important role of phytoplankton as the basis of the marine food web thereby supporting the marine life from zooplankton through fish to marine mammals, and their role in global biochemical processes, particularly in the global carbon cycling. A disruption of phytoplankton distribution will have cascading effects through the food chain with dramatic consequences for fisheries which depend on zooplankton turn depending in turn on phytoplankton.

Noteworthy results of this analysis include (1) the finding that the projected changes in plankton SR and important restructuring of open ocean phytoplankton community reveal another threat to marine ecosystem associated with ongoing global warming to anthropogenic emissions of greenhouse gases. (2) The notion that smaller warm-water diatoms and copepods species will replace larger ones at high latitudes, which will likely weaken carbon export efficiency.

The work will be of interest to marine ecologists and environmental scientists and to researchers from other fields, particularly fisheries.

However, it is also difficult to tell if there are flaws in the data analysis, interpretation and conclusions since the analyses rely entirely on software packages which are black boxes. The work would need more details to meet the expected standard.

GENERAL CONSIDERATIONS

[1] This study is comparable to a recent analysis by Ibarbalz et al. (2019), who reported uniform SST-driven latitudinal diversity gradients across trophic levels. The authors of the present study attribute the discrepancy between the findings of Ibarbalz et al and theirs to the fact that their analysis has a broader spatio-temporal coverage.

I would like to see a comparison of the results of this work to the findings of Ibarbalz et al. (2019) for areas of overlap to see if they agree.

[2] The lack of explicit specification of the SDMs models (other than simply enumerating classes of statistical and machine learning models: GLM, GAM, RF, ANN) makes it difficult to tell if the work supports the conclusions and claims. More details are required to make the work reproduced.

[3] It is also difficult to tell if there are flaws in the data analysis, interpretation and conclusions since the analyses rely entirely on software packages which are black boxes.

[4] The reported regional patterns of climate change impacts on phyto- and zoo-plankton diversity is limited on longitudinal gradient. Is there any longitudinal gradient in the regional SR patterns.

[5] The study focuses on SR which is one of the two aspects of biodiversity, the other one being relative species abundance. Can the same kind of analysis be performed with regard to species biomass rather than species occurrences? Would the data quality permit such an analysis?

SPECIFIC POINTS

[1] L77: Niche conservation is a strong assumption when considering long-term projection since species can gradually adapt to slow changes in their environment.

[2] L77: Why not consider the 4 types of SDM under all predictors and perform variable selection to retain the important predictors under each model?

[3] L80-84: The estimation of future environmental conditions of the ocean from anomalies is unclear. In particular what are the observation-based monthly climatologies? Are these data-based time series predictions for the periods of interest?

[4] L87: The statement "SR ensembles were computed as the mean sum of all species' habitat suitability patterns across all 80 possible combinations..." would suggest that SR represents habitat suitability, where SR, as defined on L87, represents species richness.

[5] L238: Please explain how were the principal components computed from changes in phyto- and zooplankton SR and associated ST rates.

MINOR POINTS

[1] L32: Replace "underlying" by "implying"

[2] L58: model-based

[3] L657-658: ...to accommodate the limited predictor availability in the future model projections...

[4] L763: Replace "which" by "with"

[5] L778 Replace "With" by "We"

COMMENTS TO THE AUTHORS OF MANUSCRIPT NCOMMS-20-37764A WITH RESPONSES FROM THE AUTHORS (IN BLUE)

1. COMMENTS FROM REVIEWER#2

I have read through the revised version of the MS of Benedetti et al, and I am fully satisfied by the way in which the authors have addressed my comments and minor concerns. Overall I find the present version of the MS further improved in terms of content and clarity. I therefore confirm my previous evaluation, i.e. this paper provides a significant contribution towards an improved understanding of the impact of climate change on the marine environment and it certainly deserves to be published in Nature Communications.

Authors: We would like to kindly thank Reviewer #2 for his/her very supportive comments and for taking the time to go through our revised manuscript. We hope the present revised version will satisfy the handling editor and all reviewers so they can read it in Nature Communications.

2. COMMENTS FROM REVIEWER#3

In this MS, the authors use SDMs to determine changes in open ocean phytoplankton species richness and community composition under a high emission scenario (RCP8.5). Their analysis is based on a total of 860 species (336 phytoplankton and 524 zooplankton) from 10 plankton functional groups. They project future habitat suitability patterns for the present (2012-2031) and the end of the century (2081-2100) according to outputs of five Earth System Models, and examine changes in alpha diversity (species richness) and beta diversity (species turnover). Overall, their results indicate an increase in phytoplankton SR by 16% over most regions for the end of century, whereas zooplankton richness is predicted to increase strongly in temperate to subpolar latitudes, but to decline in the tropics, with major changes in species turnover pointing to a major community restructuring driven by poleward range shifts.

The study touches upon a critically important subject, given the important role of phytoplankton as the basis of the marine food web thereby supporting the marine life from zooplankton through fish to marine mammals, and their role in global biochemical processes, particularly in the global carbon cycling. A disruption of phytoplankton distribution will have cascading effects through the food chain with dramatic consequences for fisheries which depend on zooplankton turn depending in turn on phytoplankton.

Noteworthy results of this analysis include (1) the finding that the projected changes in plankton SR and important restructuring of open ocean

phytoplankton community reveal another threat to marine ecosystem associated with ongoing global warming to anthropogenic emissions of greenhouse gases. (2) The notion that smaller warm-water diatoms and copepods species will replace larger ones at high latitudes, which will likely weaken carbon export efficiency.

The work will be of interest to marine ecologists and environmental scientists and to researchers from other fields, particularly fisheries.

Authors: We would like to kindly thank Reviewer #3 for stepping in as a complementary reviewer and for taking the time to read our manuscript. We appreciate her/his positive comments on our study and we are truly happy the reviewer finds it appealing to such a broad scientific community. We hope the present revised manuscript will satisfy his/her main comments below and that he/she finds it to match the standards needed for publication in Nat. Comms.

However, it is also difficult to tell if there are flaws in the data analysis, interpretation and conclusions since the analyses rely entirely on software packages which are black boxes. The work would need more details to meet the expected standard.

Authors: We are sorry to hear the reviewer did not find our study to meet the community standards regarding the description of the modelling framework we developed. In our point-by-point responses below, we highlight the various parts of our extensive Methods section and Supplementary Materials sections that address the reviewer's present comment.

We would like to assure the reviewer that every single step of our modelling framework (from biological data collection to the analyses of the species distribution models projections) was carried out by accounting for all the main sources of biases and uncertainties inherent to a niche modelling-based study. Due to the format inherent to Nat. Comms. letters, we cannot expand on the methodological choices in the main text as the focus needs to be the main findings of our study. However, we note that all of the necessary methodological choices involved in our species distribution models framework are given and supported with adequate references in our extensive Methods section. To address the reviewer's comment and further improve the transparency and reproducibility of our methods, we new filled in a standard Overview, Data, Model, Assessment and Prediction (ODMAP) protocol (<https://odmap.wsl.ch/>; see Zurell et al., 2020 - doi: 10.1111/ecog.04960, added to our list of references), i.e. a novel community standard protocol form for species distribution modelling that documents each of the steps required for model building and analyses. It summarizes the hypotheses and the goals of our ensemble SDMs approach as well as the specific parametrization of each type of SDMs, ensuring these are 100% reproducible and not just "black boxes". This way, the reviewer and future readers will have a clear and detailed overview of the aims and technicalities of our species distribution

models framework. We hope this will convince the reviewer that our study does comply with the community standards of transparency and reproducibility.

GENERAL CONSIDERATIONS

1. This study is comparable to a recent analysis by Ibarbalz et al. (2019), who reported uniform SST-driven latitudinal diversity gradients across trophic levels. The authors of the present study attribute the discrepancy between the findings of Ibarbalz et al and theirs to the fact that their analysis has a broader spatio-temporal coverage.

I would like to see a comparison of the results of this work to the findings of Ibarbalz et al. (2019) for areas of overlap to see if they agree.

Authors: *To address this insightful comment, we contacted two of the main authors of this study (Dr. Federico Ibarbalz and Pr. Chris Bowler) and requested access to their gridded spatial fields of planktonic diversity under climate change (i.e. the data underlying their maps in their Supplementary Figure S12), which were kindly provided. Hence, we were able to directly compare their model projections of copepods and photosynthetic-mixotrophic protists (labelled as "Protists (P)" in Ibarbalz et al., 2019) species diversity to our results. These two groups provide the most comparable projections to our present phytoplankton (Figure 1) and copepod (Supplementary Document S1) diversity (i.e. mean annual species richness) estimates.*

In short, the following pairs of fields were compared:

- *Contemporary mean annual phytoplankton species richness (i.e. from ensembles of habitat suitability; Figure 1d) versus their contemporary (1996-2006) estimate of Protists (P) species diversity (i.e. from Shannon index H' based on molecular measurements; their Figure S12A).*
- *Contemporary mean annual copepod species richness (Supplementary Document S1) versus their contemporary (1996-2006) estimate of copepod species diversity (i.e. from Shannon index H' based on molecular measurements; Figure S12A).*
- *Ensemble % Difference in mean annual phytoplankton species richness (2081-2100 minus 2012-2031) versus their anomalies of Protists (P) species diversity (2090-2099 minus 1996-2006; Figure S12B).*
- *Ensemble % Difference in mean annual copepod species richness (2081-2100 minus 2012-2031) versus their anomalies of copepod species diversity (2090-2099 minus 1996-2006; Figure S12B).*

Before a direct comparison could be made, we normalized both ours and their estimates of contemporary phytoplankton and copepod diversity by their respective maximum values. For each pair of variables, bivariate plots were drawn with our

own model estimates on the y axes and Spearman rank correlation coefficients (ρ) were computed. This way we evaluate the similarity of the spatial patterns in diversity between the results of Ibarbalz et al. (2019) and ours. Then, using the bivariate plots illustrating the amplitude of the future differences in diversity, we identified the regions where our projections agree or disagree on the sign of the response of protist/copepod diversity to future climate changes. Plus, by drawing the 1:1 line of these two plots, we also identified the regions where our changes in diversity are predicted to be larger or weaker than those of Ibarbalz et al. (2019), relative to the respective contemporary conditions. All the plots and the results from the correlation analyses are summarized below.

Figure caption: Comparison between our present estimates (always on the y axis) to those of Ibarbalz et al. (2019) (always on the x axis) for (a) contemporary mean annual phytoplankton/Protists (P) species diversity (species richness SR vs. Shannon diversity index H') and (c) mean difference (future decade – contemporary decade) in mean annual phytoplankton/Protists (P) species diversity. (b) same as (a) and (d) same as (c) but for copepod species diversity instead of phytoplankton/Protists (P). Each point corresponds to a 1°x1° grid cell and was colored as a function of its latitude. The dashed line represents the 1:1 line, and the dotted lines show where changes in species diversity are equal to zero.

Results from the Spearman' rank correlation tests associated to the data plotted above:

- a. $\text{Rho} = 0.852$; $\text{p-value} < 2.2 \cdot 10^{-16}$; $S = 8.019 \cdot 10^{-11}$
- b. $\text{Rho} = 0.838$; $\text{p-value} < 2.2 \cdot 10^{-16}$; $S = 8.748 \cdot 10^{-11}$
- c. $\text{Rho} = 0.248$; $\text{p-value} < 2.2 \cdot 10^{-16}$; $S = 4.064 \cdot 10^{-12}$
- d. $\text{Rho} = 0.663$; $\text{p-value} < 2.2 \cdot 10^{-16}$; $S = 8.821 \cdot 10^{-11}$

Figure caption: Comparison between our present estimates to those of Ibarbalz et al. (2019) for (a)-(b) mean difference (future - contemporary) in mean annual phytoplankton/ Protists (P) species diversity. (c)-(d) same as (a)-(b) but for copepods. Plots (a) and (c) are the same as plots

(c) and (d) in the Figure above but the points were colored according to the state of agreement between our projections and those of Ibarbalz et al. (2019) as well as the relative amplitude of the projected differences in species diversity when both estimates agree on the direction of change. Cells in red and green thus correspond to those regions where our projections disagree with those of Ibarbalz et al. (2019).

In short, we find significant correlations coefficients for four pairs of variables which indicates that our results are overall in line with those of Ibarbalz et al. (2019). Yet, the correlations vary in strength. Mean annual Protists (P)/phytoplankton and copepod species diversity display a similar patterns between the two studies (all $\text{Rho} > 0.83$), indicating that both studies find very similar latitudinal diversity gradients of species diversity for Protists (P)/phytoplankton and copepods for the contemporary ocean. Although the correlation coefficient is relatively weak ($\text{rho} = 0.248$), both studies find Protists (P)/phytoplankton diversity to increase in the future, but they strongly disagree on the response of diversity in high latitudes of the northern hemisphere (increase vs.

decrease in our case). The predicted future changes in global copepod species diversity are more similar between the two studies compared to phytoplankton, although Ibarbalz et al. (2019) found more regions where copepod diversity is likely to increase in the future.

We would like to underline that there are several major methodological differences, which makes it difficult to pinpoint the reasons behind the differences shown above. Indeed, a key issue besides the broader spatio-temporal coverage of our data, is the way plankton diversity is measured and modelled. Ibarbalz et al. (2019) derived Shannon diversity indices (H') from numbers of reads of operational taxonomic units (OTUs) obtained from high throughput 16S and 18S RNA sequencing. Therefore, it is hard to evaluate the similarity of the phytoplankton (or the "Protists (P)") community they sample compared to ours. They directly estimate species diversity at 189 sampling stations and then model it as a response variable through GAMs. Then, the authors use the latter GAMs to project species diversity in space and time as a function of sea surface temperature (SST) and chlorophyll a only. Meanwhile, we estimate species diversity as a property that emerges from stacking several hundreds of different species for which we can reliably model the global habitat suitability patterns. Furthermore, we use a broader range of model types and complexity whereas Ibarbalz et al. (2019) rely on GAMs only. Therefore, our respective approaches differ in a multitude of ways: (i) our biological observations span much broader spatial and temporal scales (several decades and all ocean basins vs. two cruises); (ii) species diversity is directly measured and modelled in their case contrary to our approach; (iii) we use a broader range of statistical models that better covers the commonly used range of algorithms and thus actually accounts for this major source of uncertainty in diversity forecasts (references #65-67); (iv) they rely on two environmental predictors (SST and chlorophyll a) to model the diversity of all the plankton groups whereas we made sure to use four different sets of predictors that span several niche dimensions adapted to each trophic level; (v) the baseline and end-of-century periods defined to compute the future environmental fields on which the statistical models are projected on differ too (10 years in their case; 20 in ours); and (vi) the Earth System Models used are not the same between our studies. This is why we believe that a direct comparison of our projections to those of Ibarbalz et al. (2019) would actually require a substantial amount of re-analysis to be fully comparable. Still, we note that our results are mostly in line with those of Ibarbalz et al. (2019) which is remarkable considering the wide differences between the two approaches. Despite these very substantial methodological differences between both studies, we believe that the comparison the reviewer suggested very worthwhile and informative, so we decided to include these additional analyses in the Supplementary Document S1 and adjusted the main text accordingly (lines 171-194).

2. The lack of explicit specification of the SDMs models (other than simply enumerating classes of statistical and machine learning models: GLM, GAM, RF, ANN) makes it difficult to tell if the work supports the conclusions and claims. More details are required to make the work reproduced.

***Authors:** We are concerned about the fact that the reviewer found our description of the SDMs to be incomplete and non-transparent. Method reproducibility is of utmost importance in science, therefore we addressed this crucial comment as follows. First, some additional details regarding the parameterization of our SDMs were added to our Methods section (see lines 785-794). Second, we filled a standard Overview, Data, Model, Assessment and Prediction (ODMAP) protocol (Zurell et al., 2020 - doi: 10.1111/ecog.04960, added to the reference list) and added it to our list of submitted Supplementary Materials (see Supplementary Document S19). All the details that could not be extensively described in the ODMAP protocol are given in our Methods section. Third, we would like to ensure the reviewer that the R codes used to perform our numerical analyses are publicly available online through the GitHub webpage of the first author (<https://github.com/benfabio>). The latter was re-organized so that all the R scripts developed for the present study are gathered in one overarching folder (<https://github.com/benfabio/Benedetti-et-al.-NCOMMS-20-37764A->). This was already the case before, but we acknowledge that the path to said folder was not straightforward as it was nested within a larger overarching one (labelled 'OVERSEE').*

We hope that these adjustments as well as the reporting of an ODMAP protocol will satisfy the reviewer and convince him/her that our Methods are reproducible. Finally, we would like to underline that all of our methodological choices were justified and referenced in our extensive Methods section. In addition, all the main biases and uncertainty sources underlying our data and results were thoroughly examined and again reported in the Methods section and/or the Supplementary Materials (but see detailed response to the reviewer's point #3 below).

3. It is also difficult to tell if there are flaws in the data analysis, interpretation and conclusions since the analyses rely entirely on software packages which are black boxes.

***Authors:** We are truly sorry the reviewer felt a lack of transparency regarding our various data analyses, how we interpret them and draw conclusions from them. However, we feel like our very extensive Methods section and our numerous Supplementary materials do cover these points as they collectively justify the various steps taken in our extensive modelling framework. Given that one of the co-authors is a pioneer in the development and validation of SDMs (Guisan & Zimmermann, 2000), and also in the development and promotion of community standards with the community (Zurell et al., 2020), and has thoroughly vetted this part of the methods and manuscripts, we are unsure as to which aspects the*

reviewer perceived to be underreported. We would also like to point out that the two other reviewers were convinced by our Methods description and did not raise this criticism. Since Nat. Comms. publishes articles in a letter format, we had to considerably shorten the description of our methods to focus on the results and the discussion. Yet, we do feel like we made a large effort to be as transparent and descriptive as possible throughout 27 pages of Methods, 18 additional materials, and the recent completion of an ODMAP protocol (cf. response to comment B above). In the Methods section, we cited the main R packages used but we also described carefully to what end and why they were used.

To assure the reviewer that we carried out the necessary steps to address the biases of the datasets and evaluate the uncertainty of our projections, we here try to summarize those steps and refer to the associated material already included in the original submission:

- *The global zooplankton species occurrence dataset was compiled to ensure we modelled those species that contribute to the composition of surface open ocean communities (section A of the Methods, Supplementary Table S7). The standards used to implement this new dataset are in line with those used in the previously published dataset of global phytoplankton occurrences (PhytoBase; Righetti et al., 2020, cited throughout the manuscript), with few adaptations to the specificities of zooplankton species. Occurrences lacking spatial and temporal coordinates were discarded. Occurrences coming from potential deep-dwelling (>500m depth) communities were discarded. Occurrences affiliated to sediment core samples were discarded (Supplementary Note S8). All species names were carefully inspected and compared to the list of accepted extant marine species names of the WoRMS, and the latter was used to homogenize species names and classification across all data sources (Supplementary Table S12).*
- *The spatial and temporal biases inherent to the present global plankton occurrence data were shown (Supplementary Figure S3; see Righetti et al., 2019 and 2020 for the phytoplankton occurrence data). To address such biases in our numerical analyses, several analyses were carried out. Pseudo-absences were drawn accounting for the biases in species distributions (section C1 of the Methods) to inform the models about the sites (i.e. monthly 1°x1° grid cell) where a species could have been sampled and recorded but was not. This was done at the several levels (total occurrence pool or group-level occurrence pool; Supplementary Figure S9; see Supplementary Material of Righetti et al., 2019) and the ensuing SDMs species richness projections were compared. The potential effects of spatial imbalance in sampling effort (i.e. historical lower sampling frequency near the equator) on our species richness projections were also examined through a thorough rarefaction analysis (Supplementary Discussion S4).*

- *The environmental predictors ultimately used to train the SDMs were selected after a very thorough selection process (section C3 of the Methods; Supplementary Methods S2) which addressed the most critical issues in niche modelling: predictors collinearity (section C3.2; Supplementary Methods S2); imbalance of the species occurrences in environmental space (section C3.1 of the Methods; Supplementary Discussion S10); reduced the quantity of predictors included in the SDMs following a conservative ratio of number of occurrences to number of predictors in order to avoid model overfitting (section C3.3; Supplementary Methods S2); and usage of four alternative sets of environmental predictors to account for the uncertainties in the choice of predictors in our SDMs projections (section C3.3; Supplementary Methods S2).*
- *The SDMs were chosen to cover the range of algorithms types and model complexity that are commonly used in the literature (see the excellent review by Merow et al., 2014 - doi: 10.1111/ecog.00845, reference #70) and which is critical to account for since SDMs choice is known to be the prime source of uncertainties in species diversity forecasts (see references #65-67). To avoid model over-fitting, one of the most common pitfalls in SDMs studies (Merow et al., 2014), we greatly reduced the number of predictors included in our SDMs to achieve a relatively high presence-to-predictors ratio of ~15 (reference #60; section C4 of the Methods). The parameters of the models were also adjusted to avoid over-fitting (section C2 of the Methods). The present SDMs and their parameterization have been previously used in numerous studies to model the ranges and the diversity patterns of various phyla, including the phytoplankton species modelled here (Righetti et al., 2019).*
- *The validity of our estimates of plankton species richness was carefully evaluated. First, species displaying poor mean evaluation metrics were discarded (section C4 of the Methods; Supplementary Figure S11). For phytoplankton, the present estimates of mean annual species richness were directly compared to the previous estimates of Righetti et al. (2019) which performed very similar analyses (cf. responses to Reviewer #1 in the earlier round of revisions). We also made sure to verify that species richness estimates derived from stacking species-level mean habitat suitability maps were similar to those estimates obtained from species-level distribution (1/0) maps derived through a probability threshold. Then, plankton functional groups-level patterns of mean annual contemporary species richness were compared and validated against to previous studies documenting global and/or regional plankton groups species diversity gradients through various observations (section E4 of the Methods; Supplementary Document S1). Finally, we further evaluated the validated of our monthly and annual species composition projections by deriving estimates of community-level size structure (median cell volume for phytoplankton and median body size for zooplankton) and comparing them*

to independent estimates (section E4 of the Methods and Supplementary Documents S17 and S18). The way we interpret our SDMs projections and species richness estimates is discussed in sections C4 and C5 of the Methods.

- The uncertainties underlying our future projections were also thoroughly examined and shown through approaches that are standard in the SDMs community. Four SDMs types trained with four alternative sets of environmental predictors were used and projected onto the predictions from five Earth System Models (ESMs), leading to no less than 80 different model members that span various ranges of future possible realizations for the future plankton communities of the global ocean (section D of the Methods; Supplementary Figures S13 and S14). Projection uncertainties were quantified and mapped (section E2 of the Methods; Supplementary Figures S13). A standard method was applied to identify those regions of the ocean where non analog conditions might emerge in the end of the century period (section E2 of the Methods; Supplementary Figures S14). Areas of model members agreement and areas of emerging non analog conditions were illustrated on our main Figure 1.

All of the steps above were thus chosen to fit the nature and bias structure of our data and carefully evaluated, rather than being carried out automatically using a “black box” R package with a generic default parameter set. Most of the abovementioned steps could have been carried out in any coding environment as they mainly consisted in the manipulation of large datasets and standard numerical analyses. We are fully aware that this is a huge amount of information to convey, but considering the space limitations inherent to the publication of scientific articles, we think we came up with the best way to thoroughly communicate how our modelling framework was developed and how we addressed each potential bias.

4. The reported regional patterns of climate change impacts on phyto- and zoo-plankton diversity is limited on longitudinal gradient. Is there any longitudinal gradient in the regional SR patterns. [Editor's note: the reviewer evidently meant "latitudinal" in the first sentence and "longitudinal" in the second.]

Authors: We did test for the existence of longitudinal patterns in our ensemble projections of mean annual phyto- and zooplankton species richness (SR) for the contemporary and future states of the global ocean. These results are summarized by the panel of zonal plots attached below which illustrates the longitudinal (1° bins) variations in phytoplankton (a,c) and zooplankton (b,c) mean annual SR (a,b) and % differences in mean annual SR (c,d). These zonal plots were drawn based on the same data used to make our main Figure 1. Overall, we find variations in mean contemporary SR and future changes in mean SR to be relatively weak across

longitudes, especially when compared to latitudinal variations. The strongest variations in mean longitudinal SR can be explained by the presence of continents that reduce the range of values and often exclude the highest values associated with tropical conditions. For instance, on the zonal plots a and b below, the dip in mean longitudinal SR observed around 20°E is due to the superimposition of the African and European continents which forbid the emergence of a tropical peak in SR. The smaller dip observed around 60°W can also be explained by the presence of the two American continents.

We compared the range (maximum - minimum value) spanned by the mean latitudinal values (Figure 1) to the range spanned by the mean longitudinal values (Figure below) for each of the variables shown below. For mean annual phytoplankton SR, the range of mean latitudinal values is 2.01 times higher than the range of the longitudinal mean values (i.e. ratio between the range of mean latitudinal values/ mean longitudinal values = 2.01). For mean annual zooplankton SR, the range of mean latitudinal values is 2.16 times higher than the range of the mean longitudinal values. For future changes in mean annual phytoplankton SR (%ΔSR), the range of mean latitudinal values is 4.49 times higher than the range of the mean longitudinal values. For future changes in mean annual zooplankton SR, the range of mean latitudinal values is 2.42 times higher than the range of the mean longitudinal values. All in all, this is why we deemed these longitudinal variations in phyto- and zooplankton mean annual SR much less interesting than the latitudinal ones and thus chose not to comment them in our manuscript.

Figure caption: Global longitudinal patterns of mean annual species richness (expressed as the sum of species-level mean annual habitat suitability) of phytoplankton (a,c) and zooplankton (b,c) in the contemporary surface ocean (a,b) and their projected changes (expressed in % difference in mean annual richness) for the 2081-2100 period under the RCP8.5 scenario (c,d).

(a)-(b) show the mean richness across all 16 ensemble members (4 species distribution models and 4 predictor pools) for the 336 phytoplankton species modelled for the contemporary ocean. (b)-(d) same as (a)-(c) but for the 524 zooplankton species modelled. The bold lines illustrate the average values per 1° longitudinal bins and the semi-transparent red ribbon illustrates the associated standard deviation. The grey points in the background represent the values from which the longitudinal averages and standard deviations were calculated (i.e. those values shown on Fig. 1).

5. The study focuses on SR which is one of the two aspects of biodiversity, the other one being relative species abundance. Can the same kind of analysis be performed with regard to species biomass rather than species occurrences? Would the data quality permit such an analysis?

Authors: We agree with the reviewer that species abundance and/or biomass would be a key parameter to account for, since not all plankton species contribute equally to community composition and total community abundance. Had data availability permitted it, we would have opted to model species-level abundances indeed. As presumed by the reviewer, we could not do so for the purposes of our study. Indeed, four main limitations related to data quality hinder us to perform such analyses for now: (i) abundance and biomass data is only available for a small subset of the species considered here, since previous collections (MAREDAT; Buitenhuis et al., 2013 - <https://doi.org/10.5194/essd-5-227-2013>) show that most observations focus on common easily observed species that constitute the majority of biomass in each functional group; (ii) merging plankton species observations on a global scale requires merging observations taken from a very broad range of sampling methods (notably plankton nets of various mesh sizes) which makes it very difficult to obtain species-level abundance measurements that are comparable across surveys/cruises, and which leads to substantial uncertainties in the derived biomass estimates (Buitenhuis et al., 2013); (iii) the two data sources (OBIS & GBIF) with the highest spatio-temporal coverage do not provide species abundance data, likely because of the reasons mentioned above, which means that species occurrence data are way more available than species abundance data, on top of being less sensitive to differences in sampling protocols; and (iv) the subset of observations with comparable abundance data (e.g. MAREDAT and COPEPOD: https://www.st.nmfs.noaa.gov/copepod/atlas/html/taxatlas_4212000.html) is mainly composed of order-level or genus-level observations and cover a much smaller range of spatio-temporal scales. For the purposes of our study, a trade-off had to be made to accommodate both the large number of species contributing to open ocean plankton communities and the broad spatio-temporal coverage needed for global diversity predictions under climate change. This is why we opted to base our models on occurrence data.

Nonetheless, we want to underline that our modelling framework based on occurrence data is able to reproduce plankton latitudinal diversity gradients that match previous observations based on various types of measurements, including

species abundance or DNA reads (Supplementary Document S2). We also derived estimates of community size structure (i.e. median cell volume and body length size) based on the monthly species-level habitat suitability that match latitudinal patterns of community size structure obtained from satellite observations or relative abundance data (Supplementary Documents S17 and S18). Therefore, we are confident that our occurrence-based model projections capture ecosystem properties that are affected by changes in species relative abundances.

On a side note, we would like to draw the reviewer's attention to the fact that our group is currently working on a large scale synthesis of plankton observations under the framework of a H2020-funded project (www.atlanteco.eu; 2020-2024). Within this effort, we are compiling species-level abundance data from multiple sources that include the main plankton abundance/biomass datasets (e.g. COPEPOD, MAREDAT) and the most recent large scale oceanographic surveys (e.g. CPR, Tara expeditions, AMT cruises etc.). Once this effort is completed, we will be able to develop a similar ensemble modelling framework to predict contemporary and future abundances and biomass for a subset of the taxa modelled here. Therefore, we will be able to perform similar analyses based on abundance/biomass data in the years to come. We would like to assure the reviewer that this is currently ongoing work and a main focus of our research agenda.

SPECIFIC POINTS

-
1. L77: Niche conservation is a strong assumption when considering long-term projection since species can gradually adapt to slow changes in their environment.

Authors: *We agree with the reviewer that this one of the main assumptions underlying our SDMs projections. Indeed, there are now a few studies documenting how some model phytoplankton species can shift their responses to thermal and pCO₂ ranges that are outside of their initial physiological limits (Lohbeck et al., 2012 - doi: 10.1038/NCEO1441; Schluter et al., 2014 - doi: 10.1038/NCLIMATE2379; Padfield et al., 2015 - doi: 10.1111/ele.12545; Schaum et al., 2017 - doi: 10.1038/s41559-017-0094; Aranguren-Gassis et al., 2019 - doi: 10.1111/ele.13378). These results support the view that single-celled photosynthetic organisms that display very fast life cycles can adapt to novel conditions and pass on physiological adaptation to the next generations, supporting the existence of niche adaptation and then evolution. However, multiple stressors (e.g. nitrogen or phosphorus limitations) can preclude adaptations to ocean warming and ocean acidification as phytoplankton have to respect trade-offs between thermal tolerance and elemental requirements (but see the very interesting study by Aranguren-Gassis et al., 2019 - doi: 10.1111/ele.13378). Therefore, the extent to which thermal adaptation occurs and can be accurately*

incorporated in SDMs for phytoplankton species, in a context of multiple stressors, is far from being well understood. Complex mechanistic evolutionary models have been developed to model evolutionary dynamics for theoretical microbes (Walworth et al., 2020 - doi/10.1073/pnas.1919332117), but those remain unsuitable for projecting the responses of entire phytoplankton communities under climate change.

Evidences of thermal adaptations in marine zooplankton species are even scarcer. Dam (2013 – doi: 10.1146/annurev-marine-121211-172229) reviewed evidences of small scale phenotypic plasticity and adaptation of zooplankton to hypoxia and harmful algal blooms but then clearly stated that evidences of long term adaptations to warming are lacking for marine zooplankton. Later on, Hinder et al. (2014 - doi: 10.1111/gcb.12387) documented how copepods in the North Atlantic Ocean showed no evidence of thermal adaptation over 50 years of ocean warming. A very recent study also showed how planktic foraminifera display strong thermal niche conservatism over the past 700 ka (Antell et al., 2021 - doi.org/10.1073/pnas.2017105118). On the opposite, there is mounting evidence that marine ectotherms are currently shifting their spatial ranges to track their suitable thermal habitat (Poloczanska et al., 2013 cited in our manuscript; Pinsky et al., 2019 - doi.org/10.1038/s41586-019-1132-4; Fredston et al., 2021 - doi: 10.1111/gcb.15614), and that such shifts are accurately captured and projected by empirical distribution models (Sunday et al., 2012 - doi: 10.1038/NCLIMATE1539; Poloczanska et al., 2013).

Therefore, based on the current state of the art, the niche conservatism assumption seems to hold for zooplankton taxa. We agree with the reviewer that this assumption is likely less robust for phytoplankton species projections, but it remains very unclear how one could adequately incorporate it into SDMs for several hundreds of different taxa.

2. L77: Why not consider the 4 types of SDM under all predictors and perform variable selection to retain the important predictors under each model?

Authors: *We admit our sentence was actually misleading here because we successively mention four types of models and then four predictors sets, which indeed might sound like each SDM type has its own specific set of predictors. Each type of SDM was indeed trained with the four different predictor set, so every model does 'see' the four possible predictors sets for the ensemble forecasting approach. To clarify, we modified the sentence as follows: "Four types of SDMs (generalized linear models, generalized additive models, artificial neural networks, and random forests [24]) were fitted to model the species' current environmental habitat suitability patterns. For each SDM we used four alternative pools of predictors".*

Predictor selection and predictor importance rankings were carefully evaluated in our study (see section C3 of the Methods and Supplementary Document S2), notably to avoid predictors collinearity and model overfitting while still capturing the

essential niche dimensions of the plankton species modelled. Ultimately, each of the four pools of predictors comprises uncorrelated variables that are known to constrain plankton distributions and that passed an exhaustive selection process based on their relative ranks of importance.

3. L80-84: The estimation of future environmental conditions of the ocean from anomalies is unclear. In particular what are the observation-based monthly climatologies? Are these data-based time series predictions for the periods of interest?

Authors: *The “observation-based monthly climatologies” correspond to those monthly climatologies of selected environmental predictors that were used to train the SDMs (see section B of the Methods for a thorough referencing of the data products that were used). We slightly modified the sentence to make this clearer. In short, the species’ monthly occurrences and pseudo-absences are matched with their corresponding closest monthly climatological values for every environmental predictor, and this constitutes the calibration set of every SDM. The latter SDMs are then projected onto the full climatological fields to obtain the contemporary estimates of global plankton diversity. The ESMs monthly projections from the 2012-2031 (baseline) and 2081-2100 (end-of-century) periods are then used to derive baseline and end-of-century monthly climatologies for the selected predictors. The difference between the end-of-century climatologies and the baseline climatologies gives us monthly model anomalies (also commonly known as “model delta” in the climate science community). These monthly model anomalies are then added to the monthly observation-based climatologies to create the final fields of future environmental conditions (see section D of the Methods). Since the SDMs are trained on in situ conditions, they cannot be directly projected onto the ESMs’ future projections. This is the standard protocol to perform climate change biodiversity scenarios (see Harris et al., 2014 - doi: 10.1002/wcc.291).*

4. L87: The statement “SR ensembles were computed as the mean sum of all species’ habitat suitability patterns across all 80 possible combinations...” would suggest that SR represents habitat suitability, where SR, as defined on L87, represents species richness.

Authors: *We admit our sentence might not have been clear enough. We do estimate species richness (for every model member) based on the sum of species habitat suitability. This approach of “stacking distribution models” is often used to estimate species richness. Often, the continuous habitat suitabilities are transformed into presence-absence (1/0) projections based on a fixed probability threshold. However, habitat suitability maps were not converted to presence-absence maps here as probabilistic outputs provide more gradual responses that should better reflect the very dynamic occupancy patterns of plankton and that are better suited than threshold approaches for our purposes (see references 75-77).*

Please see lines 985-1008 in our Methods section for a more thorough description and a discussion on how to interpret the species richness estimates stemming from our data.

The sentence was modified to clarify this point as follows: “SR ensembles were estimated from the sum of all species’ habitat suitability patterns averaged across all 80 possible combinations (hereinafter called ensemble members) of SDMs (n = 4), ESMs (n = 5) and predictor pools (n = 4)”.

5. L238: Please explain how were the principal components computed from changes in phyto- and zooplankton SR and associated ST rates.

***Authors:** We acknowledge this key piece of information could have been added in the main text. The principal components stem from a Principal Component Analysis (PCA, as described in section E3 of the Methods, lines 1196-1201). We modified this sentence to provide this key information as follows: “To investigate these regions and links, we first defined a severity index by retrieving the two first principal components of a Principal Component Analysis (PCA) that summarize 82.5% of the total variance in changes in phyto- and zooplankton SR and the associated ST rates (Methods)”.*

In short, a PCA is a multivariate ordination analysis that allows to reduce the dimensionality of large datasets. First, the input variables are standardized and rescaled to variance units (i.e. subtraction of the mean value and division by the standard deviation). Then, a symmetric covariance matrix is computed based on the covariance values of each pair of variables, which allows the PCA to summarize the correlation structure between the input variables. Finally, eigenvectors and eigenvalues are derived from the covariance matrix and are used to construct the principal components. The latter represent new variables which are linear combinations of the input variables. These combinations are orthogonal so the principal component are uncorrelated from one another and they summarize most of the information.

MINOR POINTS

1. L32: Replace “underlying” by “implying”

***Authors:** The sentence was modified accordingly.*

2. L58: model-based

***Authors:** The extra ‘s’ was removed accordingly.*

3. L657-658: ...to accommodate the limited predictor availability in the future model projections....

***Authors:** The extra ‘for’ was removed.*

4. L763: Replace “which” by “with”

***Authors:** The ‘which’ was replaced by with according to the reviewer’s comment.*

5. L778 Replace “With” by “We”

***Authors:** We corrected the sentence accordingly.*

Reviewers' Comments:

Reviewer #3:

Remarks to the Author:

Reviewer's report for "Major restructuring of marine plankton assemblages under global warming."

I enjoyed reading the revised version of the MS. The authors have addressed all my concerns with the previous version and made the necessary changes to the text. I commend the authors for taking the time to address my concerns in details, including a thorough comparison of their results to the findings of Ibarbalz et al. (2009). Even though their analyses and the analyses carried out in Ibarbalz et al. are based on different model structures and different kinds of data, it is comforting to see that the results of the two studies are broadly consistent. I am happy that the authors have realized that this kind of comparison is worthwhile, and devoted 15 lines (1150-165) and the Supplementary Document S1 to this comparison.

The authors recognize that species richness is only one aspect of biodiversity, the other one being relative species abundance. They mention that their research group is currently working on large scale synthesis of plankton observations where they compile species abundance/biomass data, which will allow them to develop a similar ensemble modeling framework to predict contemporary and future abundance/biomass for a subset of taxa involved in this study. That study will be a logical follow-up to this one, and will demonstrate whether occurrence-based models may capture ecosystem properties that affect changes in species relative abundance/biomass.

Crispin M. Mutshinda, PhD

COMMENTS TO THE AUTHORS OF MANUSCRIPT NCOMMS-20-37764B WITH RESPONSES FROM THE AUTHORS (IN BLUE)

1. COMMENTS FROM REVIEWER#3

Reviewer's report for "Major restructuring of marine plankton assemblages under global warming."

I enjoyed reading the revised version of the MS. The authors have addressed all my concerns with the previous version and made the necessary changes to the text. I commend the authors for taking the time to address my concerns in details, including a thorough comparison of their results to the findings of Ibarbalz et al. (2009). Even though their analyses and the analyses carried out in Ibarbaltz et al. are based on different model structures and different kinds of data, it is comforting to see that the results of the two studies are broadly consistent. I am happy that the authors have realized that this kind of comparison is worthwhile, and devoted 15 lines (l150-165) and the Supplementary Document S1 to this comparison.

The authors recognize that species richness is only one aspect of biodiversity, the other one being relative species abundance. They mention that their research group is currently working on large scale synthesis of plankton observations where they compile species abundance/biomass data, which will allow them to develop a similar ensemble modeling framework to predict contemporary and future abundance/biomass for a subset of taxa involved in this study. That study will be a logical follow-up to this one, and will demonstrate whether occurrence-based models may capture ecosystem properties that affect changes in species relative abundance/biomass.

Crispin M. Mutshinda, PhD

Authors: We would like to kindly thank Dr. Mutshinda again for reviewing our study and for his very supportive comments. We are truly happy he enjoyed reading our manuscript and we hope that he will be as interested by our future studies.

*On behalf of all authors.
Fabio Benedetti*